# Current Trends in Interprofessional Shared Decision-Making Programmes in Health Professions Education: A Scoping Review

Lama Sultan [1,2,*], Basim Alsaywid [3,4], Nynke De Jong [5] and Jascha De Nooijer [6]

1.  Department of Clinical Nutrition, Ministry of National Guard Health Affairs, King Abdulaziz Medical City, P.O. Box 9515, Jeddah 21423, Saudi Arabia
2.  School of Health Professions Education, Faculty of Health, Medicine and Life Sciences, Maastricht University, P.O. Box 616, 6200 MD Maastricht, The Netherlands
3.  Urology Department, King Faisal Specialist Hospital and Research Center, P.O. Box 3354, Riyadh 11211, Saudi Arabia
4.  Education and Research Skills Directory, Saudi National Institute of Health, P.O. Box 75050, Riyadh 12382, Saudi Arabia
5.  Department of Health Services Research, School of Health Professions Education, Faculty of Health, Medicine and Life Sciences, Maastricht University, P.O. Box 616, 6200 MD Maastricht, The Netherlands
6.  Department of Health Promotion, School of Health Professions Education, Faculty of Health, Medicine and Life Sciences, Maastricht University, P.O. Box 616, 6200 MD Maastricht, The Netherlands
*   Correspondence: sultanla@mngha.med.sa

**Abstract:** *Background*: Shared decision-making (SDM) is considered a patient-centred approach that requires interprofessional collaboration among healthcare professionals. Teaching interprofessional shared decision-making (IP-SDM) to students preparing for clinical practice facilitates the accomplishment of collaboration. *Objective*: This review seeks to provide an overview of current IP-SDM educational interventions with respect to their theoretical frameworks, delivery, and outcomes in healthcare. *Methods*: A scoping review was undertaken using PRISMA. Electronic databases, including OVID-MEDLINE, PubMed, OVID- EMBASE, ERIC, EBSCO-CINAHL, Cochrane Trails, APA PsycINFO, NTLTD, and MedNar, were searched for articles published between 2000 and 2020 on IP-SDM education and evaluation. Grey literature was searched for additional articles. Quality assessment and data extraction were independently completed by two reviewers, piloted on a random sample of specific articles, and revised iteratively. *Results*: A total of 63 articles met the inclusion criteria. The topics included various SDM models (26 articles) and educational frameworks and learning theories (20 articles). However, more than half of the studies did not report a theoretical framework. Students involved in the studies were postgraduates (22 articles) or undergraduates (18 articles), and 11 articles included both. The teaching incorporated active educational methods, including evaluation frameworks (18 articles) and Kirkpatrick's model (6 articles). The mean educational intervention duration was approximately 4 months. Most articles did not include summative or formative assessments. The outcomes assessed most often included collaboration and communication, clinical practice and outcome, patients' preferences, and decision-making skills. *Conclusions*: Overall, these articles demonstrate interest in teaching IP-SDM knowledge, skills, and attitudes in health professions education. However, the identified educational interventions were heterogeneous in health professionals' involvement, intervention duration, educational frameworks, SDM models, and evaluation frameworks. *Practice implications*: We need more homogeneity in both theoretical frameworks and validated measures to assess IP-SDM.

**Keywords:** health professions education; interprofessional education; shared decision-making; scoping review

## 1. Introduction

Shared decision-making (SDM) is vital in healthcare. Considered a patient-centred approach [1], it is defined as "an approach where clinicians and patients make decisions together using the best available evidence" [2] (p. 971). The core of SDM is that healthcare professionals (HCPs), patients, and/or family members collaborate in order to derive decisions from the best evidence for screening, intervention, and treatment. To ensure correspondence with a patient care plan, effective communication among HCPs, patients, and family members is crucial, involving understanding and respecting each other's perspectives [3]. It requires interprofessional collaboration (IPC) due to the complex nature of decision making for which SDM is a tool. IPC happens when two or more professions work together to achieve common goals and solve complex issues [4]. Elements of IPC include team communication, leadership, coordination, and decision making [5].

Studies have shown that SDM improves clinical outcomes [6], patients' knowledge of options [7], and patient care [8], and reduces medical costs [9]. Despite its importance, SDM has not been broadly implemented in clinical settings nor addressed in health professions education. The common barriers to SDM are lack of time, resources, and access to services [10]. Collaboration with HCPs can lead to conflicts due to hierarchal power issues [11]. Even if the physician is finally responsible in the healthcare team, all HCPs are aware of the benefits of IP-SDM in developing a better care plan.

To date, few reviews on SDM training for HCPs have been published. Légaré et al. [12] conducted a systematic review of studies to develop a conceptual model for enhancing an interprofessional approach to SDM in primary healthcare. The review shows that better results are achieved with intervention than without intervention, and when patients and HCPs are trained together. It finds an interprofessional approach to SDM important due to its impact on patient satisfaction and knowledge. It concludes that study protocols are informative for those interested in educating HCPs to improve how primary healthcare teams foster active patient participation in making coordinated health decisions. It suggests further research in developing better patient-derived measures of SDM and including patients and HCPs. Müller et al. [13] evaluated HCPs' training in SDM and analysed their evaluation strategies. They propose an evaluation framework that might be useful to structure future evaluation studies, but international agreement on a core set of outcomes is needed to improve the evidence. A scoping review by Siyam et al. [14] of interventions to promote the adoption of SDM among HCPs in clinical practice shows that SDM interventions mostly target physicians and seldom other HCPs. This review also highlights the need for SDM interventions targeting HCPs and assessing acceptability, effectiveness, and implementation.

These reviews address primary healthcare, HCP training, and SDM adoption, but none address IP-SDM interventions in health professions education. Although multiple professionals are involved in SDM, interprofessional education (IPE) is not an explicit intervention. IPE is an experience that "occurs when students from two or more professions learn about, from, and with each other" [15] (p. 10). IPE is considered a promising educational strategy that is likely to enhance the safety and quality of care, decrease medical errors, improve patient satisfaction and patient care, and enhance the knowledge and skills of professionals [16]. The reviews address some gaps on IPE learning outcomes, such as the implementation and effectiveness of IPE, the evaluation of interprofessional team outcomes rather than individual outcomes [17], the impact on patients and family members, and exploring "how team members handle psychological obstacles" including attitudes and practices in providing IP-SDM [18], and the lack of validated outcome measures [12]. Given these findings, our scoping review aims to provide an overview of current interventions with respect to the theoretical frameworks, delivery methods, and outcomes of these programmes. We include both peer-reviewed and grey literature to increase the comprehensiveness of our review.

## 2. Methods

We followed Arksey and O'Malley's framework [19] for conducting a scoping review and the PRISMA-ScR for reporting items.

### 2.1. Research Questions

**Research Question 1 (RQ1):** *What are the components of IP-SDM educational interventions and which theories are they based on?*

**Research Question 2 (RQ2):** *What are the current delivery methods of IP-SDM educational interventions?*

**Research Question 3 (RQ3):** *What are the outcomes of IP-SDM educational interventions and how are these assessed?*

### 2.2. Search Strategy

We included the following electronic databases, hand searches, and grey literature for papers published between 1 January 2000 and 28 September 2020: OVID-MEDLINE, PubMed, OVID-EMBASE, ERIC, EBSCO-CINAHL, Cochrane Trails, APA PsycINFO, NTLTD, and MedNar. Search terms (MeSH headings or keywords) in title or abstract were derived from 2 main concepts: interprofessional education and shared decision-making *(Interprofessional education **OR** inter-professional education **OR** IPE **OR** interprofessional practice **OR** interprofessional competencies **OR** interprofessional collaboration **OR** IPC **OR** interdisciplinary team **OR** collaborative practice **OR** collaborative learning **OR** team learning **OR** shared learning **OR** healthcare professions **OR** healthcare professionals **OR** health professions **OR** health professionals)* **AND** *(shared decision-making **OR** decision-making **OR** interprofessional shared decision-making model **OR** interprofessional shared decision-making **OR** SDM **OR** IP-SDM)* **NOT** *(business **OR** economics **OR** managers **OR** management).* We hand-searched reference lists of all selected articles to locate any potentially relevant records that had not been obtained in the first search. We also performed a search in Opengrey and Grey Literature in the Netherlands (GLIN).

### 2.3. Article Selection

The process of article selection was blinded in terms of authors, years of publication, and journals. After the individual inclusion and exclusion processes, the judgements of the two reviewers were compared. Eligibility screening was a three-step process. Titles were first screened by two reviewers (L.S. and B.A.) independently. Second, the same reviewers screened the abstracts, and third, the same reviewers independently conducted full-text screening for eligibility criteria. The reasons for exclusion are noted in (Table S1).

#### 2.3.1. Eligibility Criteria

Studies from peer-reviewed literature published between 2000 and 2020 were included due to the evolution of the topic in the 2000s, in addition to the shift toward interprofessional healthcare teams and patient-centred care in that period. Because of limited resources for translation, only studies that were written in the English language were included. Studies were included that focused on students in under- and postgraduate HCP education, including interns, residents, and fellows. Interventions included at least two different HCPs and SDM and addressed knowledge, attitudes, and/or skills. With regard to the outcomes, studies were included if they reported on summative or formative evaluations of HCPs' education in SDM as well as outcomes that had an impact on patient care and/or the healthcare system. Other eligible studies used reviews, quantitative, qualitative, and/or mixed-methods designs, or were intervention studies or descriptive studies. Eligible grey literature included relevant studies that targeted SDM and HCPs, reports, and conference abstracts.

Studies were excluded if they focused on students in foundation year or on senior healthcare team members, or when interventions took place in work-based learning in

healthcare settings excluding internship, residency, and fellowship. Studies unrelated to HCPs were also excluded, as were personal opinions and letters to the editor, as well as non-English articles and articles without full text.

### 2.3.2. Quality Assessment

The quality of each article that met the study inclusion criteria was assessed with 11 quality indicators for selection developed by Buckley et al. [20]. These related to the research question, study subjects, data collection method, completeness of data, control confounding, analysis of results, conclusions, reproducibility, prospective, ethical issues, and triangulation. Higher-quality studies were considered which met a minimum of 7 of these 11 indicators (Table S2).

### 2.4. Charting the Data

A data abstraction form was drafted, discussed with all co-authors, and tested independently by two reviewers (L.S. and B.A.) on a random sample of articles and revised iteratively by the whole team. The extracted variables are presented in Tables 1–3. General information on the study includes: study period, country, study population, sample size, study design, methodology, SDM definition, data collection methods, conclusion, and recommendations (Table 1). The IP-SDM intervention includes: disease, clinical area, health professionals' involvement, undergraduate or postgraduate, patient/family member involvement, type of decision, educational framework, teaching method, focuses on knowledge, attitudes, and/or skills, intervention duration, SDM model and components (Table 2). Outcomes includes: evaluation framework, type of outcome, SDM measures and instruments, summative and/or formative assessment, and results (Table 3).

**Table 1.** General information on included articles.

| Ref No. | Author (s), Year of Publication | Title | Study Period | Country | Study Population and Sample Size (If Applicable) | Study Design | Methods/Methodology | SDM Definition | Data Collection Methods | Conclusion | Recommendations |
|---------|-------------------------------|-------|-------------|---------|------------------------------------------------|-------------|---------------------|---------------|------------------------|-----------|----------------|
| [12] | Légaré et al., 2008 | *Advancing theories, models and measurement for an interprofessional approach to shared decision-making in primary care: a study protocol* | Not reported | Canada | (n = 70) experts in the field | Systematic review | Based on conceptual model and a set of measurement tools used to enhance an interprofessional approach to SDM in primary healthcare and pilot-tested with key stakeholders and primary healthcare teams. | A process by which a healthcare choice is made by practitioners together with the patient. | Face-to-face team meeting, interviews, and focus groups | This study protocol is informative for researchers interested in designing and/or conducting future studies and educating health professionals to improve how primary healthcare teams foster active participation of patients in making health decisions. | The need to foster a more coordinated interprofessional effort for implementing SDM in clinical practice. |
| [13] | Müller et al., 2019 | *Strategies to evaluate healthcare provider trainings in shared decision-making (SDM): a systematic review of evaluation studies* | Not reported | Germany | Not reported | Systematic review | Systematic review of studies evaluating healthcare provider trainings in SDM to analyse their evaluation strategies. | The patient and at least one clinician share information and values, deliberate the next step, and arrive at a jointly made decision. | Not reported | Strategies to evaluate HCP trainings in SDM varied largely. | The proposed evaluation framework maybe useful to structure future evaluation studies, but international agreement on a core set of outcomes is needed to improve evidence. |
| [18] | Col et al., 2011 | *Interprofessional education about shared decision-making for patients in primary care settings* | Not reported | United Kingdom | Not reported | Framework development | A two-part review highlights key elements for consideration in planning and implementing interprofessional educational interventions. | Decision making in preventive care. | Not reported | A framework for educators to construct their own teaching models following adult learning. | Understanding the concept of SDM; acquiring relevant communication skills to facilitate SDM; understanding professional values/sensitivities; understanding the roles of different professions within the relevant primary care group; and acquiring relevant skills to implement SDM. |
| [21] | Kryworuchko et al., 2013 | *Interventions for Shared Decision-Making About Life Support in the Intensive Care Unit: A Systematic Review* | Not reported | Canada | Not reported | Systematic review | A systematic review of randomized controlled trials of SDM interventions for the decision about using life support, limiting the use of life support, or withdrawing life support for hospitalized patients. | A process where healthcare professionals engage the patient and their family or surrogate decision-maker in the essential elements of the SDM process. | Not reported | Emerging evidence to guide clinical practice suggests that having someone on the interprofessional team assigned to the role of facilitating communication of the essential elements of SDM improves health outcomes. | Interventions that include essential elements of SDM need to be more thoroughly evaluated in order to determine their effectiveness and health impact and to guide clinical practice. |
| [22] | Orchard et al., 2012 | *Assessment of Interprofessional Team Collaboration Scale (AITCS): Development and Testing of the Instrument* | Not reported | Canada | (n = 125) practitioners from 7 healthcare teams and (n = 24) IPE experts | Diagnostic study | The characteristics of collaboration used to generate items related to each element. Scale items represent the 4 elements that are considered key to collaborative practice. | A process in which the patient and providers consider outcome probabilities and patients' preferences and reach a healthcare decision based on mutual agreement. | Survey | The AITCS can be applied to continuing professional education interventions to determine change over time. | Further test and retest reliability and longitudinal study application are needed. |
| [23] | Thomson et al., 2017 | *Making Decisions Better: an evaluation of an educational Intervention* | Not reported | United Kingdom | (n = 85) primary care health professionals | Pre-intervention and post-intervention. | Three groups of primary care health professionals completed questionnaires using Likert scales to assess strength of agreement with decision-making statements. | Multiple complex skills, including information mastery, numeracy, communication of risks and benefits using a variety of techniques, and the interplay of two peoples' cognitive and affective biases. | Questionnaire | Participation in the learning sessions significantly improved self-reported understanding of decision-making processes and application to clinical practice. | Further research should be undertaken to continue to build the evidence base for the explicit impact of decision-making teaching on evidence-based and individualized care. |
| [24] | Waldron et al., 2016 | *Development of a video-based education and process change intervention to improve advance cardiopulmonary resuscitation decision-making* | 13 months | Australia | 2 focus groups, (n = 8) consultants and (n = 10) junior doctors | Literature review | Utilize a framework to develop an intervention to improve advance CPR decision making. | A discussion with the patient that should be used to reach a common understanding about the medical treatment plan. | Focus groups | Approaches were developed to address physician and systemic barriers to advance CPR decision making and documentation. | Implementation and evaluation across hospital settings is required to examine utility and determine effect on quality of care. |

**Table 1.** *Cont.*

| Ref No. | Author (s), Year of Publication | Title | Study Period | Country | Study Population and Sample Size (If Applicable) | Study Design | Methods/Methodology | SDM Definition | Data Collection Methods | Conclusion | Recommendations |
|---|---|---|---|---|---|---|---|---|---|---|---|
| [25] | Sangaleti et al., 2017 | *Experiences and shared meaning of teamwork and interprofessional collaboration among health care professionals in primary health care settings: a systematic review* | Not reported | Brazil | Not reported | Systematic review | A three-step search strategy was utilized. Ten databases were searched for papers published from 1980 to June 2015. | Not reported | Not reported | This review has identified possible actions that could improve implementation of teamwork and IPC in primary healthcare. | Not reported |
| [26] | Nguyễn et al., 2019 | *Conventional and Complementary Medicine Health Care Practitioners' Perspectives on Interprofessional Communication: A Qualitative Rapid Review* | 3 months | Australia | Not reported | Qualitative rapid literature review | Six databases were searched to identify original research and systematic reviews published since 2009. | "Sharing a philosophy of care and a common understanding pertaining to scope of practice and area of expertise" "Agreement among the practitioners of a shared vision, open-minded culture, credible supporters, suitable facilities and confidence in the clinical competency of the other practitioners" | Surveys, questionnaires, semi-structured interviews | IPC within and between conventional and complementary HCP is impacted by inter-related factors. | A diverse range of initiatives that facilitate interprofessional learning and collaboration are required to facilitate IPC and help overcome medical dominance and interprofessional cultural divides. |
| [27] | Shiao et al., 2019 | *Creation of nurse-specific integrated interprofessional collaboration and team efficiency scenario/video improves trainees' attitudes and performances* | Not reported | Taiwan | (n = 36) nursing trainees, (n = 24) standardized partners | Prospective study | Mock simulation with two scenarios was held as pre-intervention IPC-TE assessment. Basic and advanced workshops were arranged for teams of intervention groups for creation of discipline-specific scenario and video. | Not reported | Survey | The implementation of a scenario creation-based training resulted in additional improvement in trainee IPC and TE behaviours and attitudes. | Future research can explore the impacts of this interventional program on clinical practice and long-lasting dynamics among nursing teams and other professional teams. |
| [28] | Voogdt-Pruis et al., 2019 | *Improvement of shared decision-making in integrated stroke care: a before and after evaluation using a questionnaire survey* | 7 months | Netherlands | (n = 25) healthcare professionals | Before and after evaluation study | The SDM implementation programme consisted of training for healthcare professional, tailored support, development of decision aids, and a social map of local stroke care. | An approach where clinicians and patients share the best available evidence when faced with the task of making decisions, and where patients are supported to consider options, to achieve informed preferences. | Questionnaire and in-depth interviews | The study indicated its feasibility to implement SDM in integrated stroke care. | Special attention should be given to the following activities: (1) the appointment of knowledge brokers, (2) agreements between HCPs on roles and responsibilities, (3) the timely investigation of patient's preferences in the care process through discussions in a multidisciplinary meeting. |
| [29] | Légaré et al., 2011 | *Interprofessionalism and shared decision-making in primary care: a stepwise approach towards a new model* | 7 months | Canada | (n = 4) nurses, (n = 3) physicians, (n = 1) dietician, (n = 1) psychologist, (n = 1) anthropologist, and (n = 1) community health specialist | Model development | Participants were divided into 3 small interdisciplinary groups and were charged with using the blocks to develop and draw the figure of a new conceptual model in primary care. | A process by which a healthcare choice is made by a practitioner together with the patient and is said to be the crux of patient-centred care. | Questionnaire | The new IP-SDM model for primary care has the potential to unify the process of SDM in different healthcare system settings and with different health professionals. | It is important to identify factors that could affect the model's implementation in primary healthcare practice, education, and applied health services research. |
| [30] | McLaughlin et al., 2014 | *Rational and Experiential Decision-Making Preferences of Third-Year Student Pharmacists* | Not reported | United States of America | (n = 114) pharmacist students | Mixed-method study | To collect data about student pharmacist decision-making styles. | A complex process that can vary based on individual, social, and context-specific influences. | Electronic survey | Student pharmacists favoured rational decision making over experiential decision making, which was similar to results of studies performed of other health professions. | This study suggested that there are 2 independent modes of processing that operate simultaneously and sequentially during decision making. |
| [31] | Chung et al., 2016 | *Educational interventions to train healthcare professionals in end-of-life communication: a systematic review and meta-analysis* | Not reported | Canada | Not reported | Systematic review | MEDLINE, Embase, CINAHL, ERIC, and the Cochrane Central Register of Controlled Trials were searched. | Interventions designed solely for information-sharing. | Not reported | Very-low- to low-quality evidence suggests that end-of-life communication training may improve healthcare professionals' self-efficacy, knowledge, and EoL communication scores compared to usual teaching. | Further studies comparing two active educational interventions are recommended with a continued focus on contextually relevant high-level outcomes. |
| [32] | Diouf et al., 2016 | *Training health professionals in shared decision-making: Update of an international environmental scan* | 24 months | Canada | Not reported | Systematic review | Two systematic reviews were shared for SDM training programs targeting health professionals produced from 2011 to 2015. | A decision-making process jointly shared by patients and their healthcare providers. | Not reported | SDM training programs still vary widely. Most still focus on the single provider/patient dyad and few are evaluated. | Integration of SDM training into the mandatory academic curricula of health professionals to ensure a better dissemination of interprofessional SDM. |

**Table 1.** *Cont.*

| Ref No. | Author (s), Year of Publication | Title | Study Period | Country | Study Population and Sample Size (If Applicable) | Study Design | Methods/Methodology | SDM Definition | Data Collection Methods | Conclusion | Recommendations |
|---|---|---|---|---|---|---|---|---|---|---|---|
| [33] | Johnsen et al., 2016 | *Teaching clinical reasoning and decision-making skills to nursing students: Design, development, and usability evaluation of a serious game* | Not reported | United Kingdom | (n = 6) nursing students and faculty members | Prototype development | Unified framework of usability called TURF (Task, User, Representation, and Function) and SG theory were employed to ensure a user-centred design. | Not reported | Questionnaire and individual interviews | The SG was perceived as being useful, usable, and satisfying. | The achievement of the desired functionality and the minimization of user–computer interface issues emphasize the importance of conducting a usability evaluation during the SG development process. |
| [34] | Kryworuchko et al., 2016 | *Factors influencing communication and decision-making about life-sustaining technology during serious illness: a qualitative study* | 12 months | Canada | (n = 30) healthcare professionals | Qualitative study | Used Flanagan's critical incident technique (CIT) and interpretive description of open-ended interviews. | The integration of information about options with the patient's values and preferences. | Open-ended in-depth individual interviews | A focus on more meaningful and productive dialogue with patients and families may improve decisions about life-sustaining technology. | Work is needed to acknowledge and support the non-curative role of healthcare and build capacity for the interprofessional team to engage in effective decision-making discussions. |
| [35] | Lestari et al., 2016 | *Understanding students' readiness for interprofessional learning in an Asian context: a mixed-methods study* | Not reported | Indonesia | (n = 470) students from (medicine, nursing, midwifery, and dentistry) | Explanatory, sequential mixed-methods design | Collected quantitative data and the results of the questionnaire were then used as input for the qualitative data collection consisting of mono-professional focus group discussions. | Not reported | Mono-professional focus group discussions | Students were generally favourable to IPE opportunity that offered to them interprofessional leadership, collaboration, and communication skills. | The present study revealed several important reasons underlying students' positive and negative perceptions of IPE implementation which may be addressed during the interprofessional learning process. |
| [36] | Lütgendorf-Caucig et al., 2017 | *Vienna Summer School on Oncology: how to teach clinical decision-making in a multidisciplinary environment* | 7 days | Austria | (n = 30) medical students | Educational approach | The program is comprised of two parts: clinical (T1) and research (T2). | Clinical decision making | Questionnaire | Clinical decision making should proceed based on the results of prototypic case-based-derived knowledge supporting associative and procedural learning processes. | Students should be prepared for multidisciplinary teaching in under- and postgraduate cancer education. |
| [37] | Légaré et al., 2018 | *Interventions for increasing the use of shared decision-making by healthcare professionals (Review)* | Not reported | Canada | Not reported | Review | CENTRAL, MEDLINE, Embase, and five other databases were searched on 15 June 2017 and two clinical trials registries and proceedings of relevant conferences. | A process by which a healthcare choice is made by the patient, significant others, or both with one or more healthcare professionals. | Not reported | Studies in this field of research are no different from those in other fields in that their methods may be inadequate; they may be too small; many fail to deal adequately with bias; and most are not replicated. | More and better research is required to strengthen the certainty of the evidence. |
| [38] | Diendéré et al., 2019 | *How often do both core competencies of shared decision-making occur in family medicine teaching clinics?* | 12 months | Canada | (n = 71) health professionals and (n = 238) patients | Qualitative and quantitative cross-sectional study | Collected a convenience sample of 250 visits in primary care, approximately 50 visits per site, considering both the need for a range of primary care consultations and the feasibility of the study. | The collaborative process by which health professionals and patients partner to make evidence-informed health decisions that reflect what matters to patients and their families. | Questionnaire | Health professionals in family medicine are making an effort to engage patients in shared decision making in routine daily practice. | The greatest potential for improvement might lie in value clarification; that is, discussing what matters to patients and families. |
| [39] | Noguera et al., 2019 | *Student's Inventory of Professionalism (SIP): A Tool to Assess Attitudes towards Professional Development Based on Palliative Care Undergraduate Education* | Not reported | Spain | (n = 300) medical students | Sequential exploratory strategy mixed method | The inventory is built based on the themes that emerged from the analysis of four qualitative studies about nursing and medical students' perceptions related to palliative care teaching interventions. | Helps medical students address several competencies related to being patient-centred and empathic. | Survey | This new inventory is grounded on students' palliative care teaching experiences and seems to be valid to assess students' professional development. | Including sociodemographic variables in future studies would allow to study which other personal and cultural factors influence professionalism learning. |
| [40] | Rajendran et al., 2019 | *Shared decision-making by United Kingdom osteopathic students: an observational study using the OPTION-12 Instrument* | Not reported | United Kingdom | (n = 30) medical students | Instrument validation | The use of reliable and validated OPTION-12 (O12) instrument to calculate a score that reflected the degree of SDM utility. | An approach where clinicians and patients make decisions together using the best available evidence. | Interviews | Students in this study did not practice competent SDM behaviours. | Effective educational strategies are required to ensure SDM behaviours reach competent levels. |
| [41] | Allaire et al., 2012 | *What Motivates Family Physicians to Participate in Training Programs in Shared Decision-Making?* | Not reported | Canada | (n = 39) family physicians | Pilot randomized trial | Small, interactive group workshops at each family medicine group. | The physician and the patient make a decision together based on the best available evidence and on the patient's values and preferences, without discounting those of the physician. | Questionnaire and focus groups | Findings from this study cannot be generalized to the larger population of physicians, and additional research is needed to refine the understanding of factors influencing FPs' participation in CPD programs in SDM. | CPD developers should make the program interesting, enjoyable, and professionally stimulating. |

**Table 1.** *Cont.*

| Ref No. | Author (s), Year of Publication | Title | Study Period | Country | Study Population and Sample Size (If Applicable) | Study Design | Methods/Methodology | SDM Definition | Data Collection Methods | Conclusion | Recommendations |
|---|---|---|---|---|---|---|---|---|---|---|---|
| [42] | Beitinger et al., 2014 | *Trends and perspectives of shared decision-making in schizophrenia and related disorders* | Not reported | Germany | Not reported | Narrative review, systematic review | Narrative review of important studies on SDM in the years before 2012 and a systematic review for the time period May 2012–October 2013. | A model of how doctors and patients make medical decisions, which is seen as very applicable to mental health. | Questionnaire and interviews | SDM in mental health is complex, takes time, and involves more than just two participants; patients' lack of decisional capacity is seen as the major barrier. | Healthcare professionals need more training in how to deal with difficult decisional situations. |
| [43] | Allen et al., 2020 | *Implementing a shared decision-making and cognitive strategy-based intervention: Knowledge user perspectives and recommendations* | Not reported | Canada | (n = 10) clinicians | Exploratory qualitative research design | Cognitive strategy-based intervention approach. | A person-centred process in which clinicians and patients collaborate to make decisions about assessments, treatment goals, and subsequent evidence-based treatment plans. | Semi-structured focus group | This study is based on a real-world implementation of an SDM-based intervention from the perspective of individual allied health professionals and interprofessional stroke rehabilitation teams. | Facilitators should lay out a framework for training, communication, and implementation that is structured but still provides flexibility for iterative learning and active problem-solving within the relevant practice context. |
| [44] | Kienlin et al., 2020 | *Ready for shared decision-making: Pretesting a training module for health professionals on sharing decisions with their patients* | 5 months | Norway | (n = 429) nurses, physicians, and health professional students | Descriptive mixed-methods study | The training was provided as two different applications (module AB (introduction and SDM-basics) and module ABC (introduction, SDM-basics, and interactive training)) with differing learning objectives, extent of interactivity, and duration (1 vs. 2 h). | A best practice approach for decision-making communication about health-related issues. | Questionnaires and focus group | The two SDM training modules met the basic requirements for use in a broader SDM implementation strategy and can even improve knowledge. | Findings to improve the education suggest higher emphasis on interprofessional teaching methods. |
| [45] | Keshmiri et al., 2020 | *The effect of interprofessional education on healthcare providers' intentions to engage in interprofessional shared decision-making: Perspectives from the theory of planned behaviour* | Not reported | Iran | (n = 113) ED residents and nurses | Sequential explanatory mixed method | The intervention group was exposed to case-based learning sessions conducted by applying interprofessional strategies. Then, they were assessed before and 2 weeks after the intervention by a questionnaire designed based on the theory of planned behaviour. | Collaboration to make decisions about assessments and treatment goals. | Questionnaire, semi-structured individual interviews | The major findings of the current study indicated that IPE could significantly improve the learners' intentions to engage in IP-SDM. | There is a need to develop the infrastructure of IP-SDM at different elements such as providers, administers, consumers, and contextual factors. |
| [46] | Reed et al., 2017 | *Linking Essential Learning Outcomes and Interprofessional Collaborative Practice Competency in Health Science Undergraduates* | 4 months | United States of America | (n = 94) medical students | Mixed methods | Several ethical decision-making models were presented, and student groups were required to use one to work through the ethical issues and come to a decision. | Articulate the impact of personal values and professional ethics in healthcare decision making. | Group presentation, individual scholarly paper | Results were as expected given students' level of progression in the program and the university. | The strategy has potential for use in assessing a variety of Student Learning Outcomes if closely linked with course, program, and college outcomes. |
| [47] | Wainwright et al., 2011 | *Factors That Influence the Clinical Decision-Making of Novice and Experienced Physical Therapists* | Not reported | United States of America | (n = 3) clinicians | Qualitative research methods using grounded theory | Three participant pairs (each pair consisting of one novice and one experienced physical therapist). Case summaries of each participant provided the basis for within- and cross-case analysis. | A process including skills such as critical thinking and problem solving, which are essential to making appropriate decisions and taking action for the effective care of patients. | Interview | The results of the study may be used by educators and employers to develop and structure learning experiences and mentoring opportunities for students and novice learners. | The results of the present study may be used by academic and clinical educators to develop and structure learning experiences to facilitate CDM and reflection for novice clinicians or students. |
| [48] | Hansen et al., 2012 | *Life-Sustaining Treatment Decisions in the ICU for Patients with ESLD: A Prospective Investigation* | 14 months | United States of America | (n = 6) patients, (n = 19) family members, and (n = 122) health professionals | Prospective, multiple case design | Case studies began within 24–48 h of ICU admission and ended when LSTs were withheld or withdrawn, or when a patient died or was transferred out of the ICU. | Process by providing information about minor decisions and assessing families' understanding of treatments. | Bedside observation, semi-structured interviews, medical record reviews, quantitative survey. | Sub themes described why patients and family members may not fully understand or comprehend the LST decision-making process. | Further research is needed to develop interventions that target patients, family members, and healthcare professionals. |
| [49] | Thompson et al., 2013 | *An agenda for clinical decision-making and judgement in nursing research and education* | Not reported | United Kingdom | Not reported | Review | The paper presents nine unanswered questions that researchers and educators might like to consider as a potential agenda for the future of research into this important area of nursing practice, training, and development. | The act of choosing between alternatives. | Not reported | The paper highlights the role of decisions and judgements made by nurses in improving quality in healthcare systems. | The real methodological, theoretical, and empirical advances will come from researchers and educators grappling with answering these questions. |

**Table 1.** *Cont.*

| Ref No. | Author (s), Year of Publication | Title | Study Period | Country | Study Population and Sample Size (If Applicable) | Study Design | Methods/Methodology | SDM Definition | Data Collection Methods | Conclusion | Recommendations |
|---------|---------------------------------|-------|--------------|---------|------------------------------------------------|--------------|---------------------|----------------|-------------------------|------------|-----------------|
| [50] | Gigue're et al., 2012 | *Development of PRIDe: A tool to assess physicians' preference of role in clinical decision-making* | 6 months | Canada | (n = 39) family physicians | Pilot clustered randomized clinical trial | Evaluated the effectiveness of DECISION+. | When a doctor and a patient engage in a joint decisional process that is informed by the best scientific evidence on the harms and benefits of the relevant interventions and by the patient's values and preferences. | Questionnaire | SDM training shows promising results, and the next step is to develop more clinical vignettes followed by questions inspired from this analysis. | The PRIDe instrument can be used in the assessment of health professionals' attitude towards SDM after training in SDM. Additional research is needed to evaluate its validity before it can be recommended for use. |
| [51] | Körner et al., 2012 | *Interprofessional SDM train-the-trainer programme "Fit for SDM": provider satisfaction and impact on participation* | Not reported | Germany | (n = 15) patients | Not reported | In step 1 the university project team trained the providers in executive positions in the clinics as trainers, who then in step 2 trained their staff in the healthcare team. | Not reported | Questionnaire | This is the first interprofessional SDM train-the-trainer program in Germany to bridge interprofessionalism and SDM. It was implemented successfully and evaluated positively. | Establishing IP- SDM training programs should be encouraged for all healthcare professionals. |
| [52] | Sheridan et al., 2012 | *Shared decision-making for prostate cancer screening: the results of a combined analysis of 2 practice-based randomized controlled trials* | 13 months | United States of America | (n = 36) physicians | Two separate randomized controlled trials | Two separate randomized controlled trials of the same prostate cancer intervention. | A process in which patients are involved as active partners in clinical decisions. | Survey | SDM interventions can increase men's knowledge, alter their perceptions of prostate cancer screening, and reduce actual screening. However, they may not guarantee an increase in shared decisions. | More work is needed to determine the added value of a shared decision. |
| [53] | Yu et al., 2015 | *Impact of an interprofessional shared decision-making and goal-setting decision aid for patients with diabetes on decisional conflict—study protocol for a randomized controlled trial* | 12 months | Canada | (n = 40) patients with physician 1:1 ratio | Randomized controlled trial | The first step is a provider-directed implementation only; the second (after a 6-month delay) involves both provider- and patient-directed implementation. | Is the process whereby two or more healthcare professionals are involved in making the decision with the patient. | Individual semi structured interview | An individualized approach to patients with multiple chronic conditions using SDM and goal setting is a desirable strategy for achieving guideline-concordant treatment in a patient-centred fashion. | This trial will provide insights regarding strategies for the routine implementation of such interventions in clinical practice, and it will offer an assessment of the impact of this approach. |
| [54] | Giguère et al., 2018 | *Tailoring and evaluating an intervention to improve shared decision-making among seniors with dementia, their caregivers, and healthcare providers: study protocol for a randomized controlled trial* | Not reported | Canada | (n = 49) clinicians and (n = 27) caregivers | Two-armed, clustered randomized trial | Two phases: (1) design and tailor the intervention; and (2) implement and evaluate. | Proposes that clinicians and patients collaborate to make joint decisions based on the best evidence. | Interview approaches, questionnaires and audio-recorded discussions | The intervention empowered patients and their caregivers in their healthcare by fostering their participation as partners during the decision-making process. | Not reported |
| [55] | Hendricks- Ferguson et al., 2018 | *Undergraduate students' perspectives of healthcare professionals' use of shared decision-making skills* | Not reported | United States of America | (n = 42) students | Exploratory qualitative approach | Data consisted of student responses in a course reflection assignment that captured their perspectives about recommended SDM responses by HCPs. | Small-group discussions | Student reflection assignments | IPE and healthcare students can develop an understanding of SDM and ethical principles related to PCC. | Not reported |
| [56] | Arenth et al., 2019 | *Teaching the Skill of Shared Decision-Making Utilizing a Novel Online Curriculum: a Blinded Randomized Controlled Pilot Study (S803)* | Not reported | United States of America | Not reported | Not reported | The intervention group received a brief online curriculum aimed at teaching the skill of shared decision making. Participants from both groups then repeated the same simulation and were reassessed. | Conversations | Video recorded | An easily accessible educational intervention in the form of an online module format is an effective way of teaching these behaviours. | SDM behaviours in non-palliative care paediatric providers can be significantly improved by access to online educational modules. |

**Table 1.** *Cont.*

| Ref No. | Author (s), Year of Publication | Title | Study Period | Country | Study Population and Sample Size (If Applicable) | Study Design | Methods/Methodology | SDM Definition | Data Collection Methods | Conclusion | Recommendations |
|---|---|---|---|---|---|---|---|---|---|---|---|
| [57] | Hagoel et al., 2011 | *Interprofessional education about decision support for patients across cultures* | Not reported | United States of America | Not reported | Curricula design | The literature on cultural competency and DS offers guidance on the objectives, competencies, and teaching strategies for an IP cross-cultural DS curriculum. | The potential to create misunderstandings and barriers among providers and between them and patients. | Videos of simulated cross-cultural, self-reflection, cross-cultural interactions with simulated patients, role play, observation | The literature on cultural competency and DS offers guidance on the objectives, competencies, and teaching strategies for an IP cross-cultural DS curriculum. | These topics are fertile ground for future research efforts in both education and healthcare, with findings that would support the refinement of decision aids and the movement of culturally competent DS into IP curricula and practice. |
| [58] | Lown et al., 2011 | *Continuing professional development for interprofessional teams supporting patients in healthcare decision-making* | Not reported | United States of America | Not reported | Curriculum development | Modification of the six-step approach to curriculum development advocated by Kern et al. to develop the model. | A complex process in which mutual influence, context, preferences, values, and information are shared in both the process and decision outcomes. | Questionnaire, open-ended questions, and semi-structured interviews | This model aligns curricular goals, objectives, educational strategies, and evaluation instrument strategies with desired learning and organizational outcomes. | Educational leaders and researchers can institutionalize such a model. |
| [59] | Neville et al., 2013 | *Team decision-making: design, implementation and evaluation of an interprofessional education activity for undergraduate health science students* | 6 months | Australia | (n = 33) nursing students, (n = 10), midwifery students, (n = 18) medical students | Cross-sectional study | All students were informed about this IPE program during an introductory lecture, which provided the evidence for the value of team decision making. The following week, students were allocated to an interprofessional mixed group that assessed the key issues. | Not reported | Questionnaire | Design, implementation, and evaluation of an IPE, team decision-making activity were reported. | This study contributed to the development of an innovative curriculum activity, which provided the opportunity for health science students to participate effectively in team decision making with the purpose of achieving better health outcomes. |
| [60] | Thistlethwaite et al., 2016 | *Introducing the individual Teamwork Observation and Feedback Tool (iTOFT): Development and description of a new interprofessional teamwork measure* | Not reported | Australia | Not reported | Not reported | Not reported | Not reported | Not reported | The advanced version is for senior students and junior health professionals and has 10 observable behaviours under four headings: "shared decision making", "working in a team", "leadership", and "patient safety". | Further testing is required to focus on its validity and educational impact. |
| [61] | Elwyn et al., 2017 | *A three-talk model for shared decision-making: multistage consultation process* | 12 months | United States of America | (n = 488) clinicians from 6 specialties | Multistage consultation process | Step 1: key informant commentary on revised model, Step 2: distribution of online survey to wider communities of interest, Step 3: review by medically qualified clinicians in six clinical specialties. | A process in which decisions are made in a collaborative way, where trustworthy information is provided in accessible set of formats about a set of options. | Survey | The revised model conveys the core principles of shared decision making by proposing easy-to-remember conversational steps to facilitate the use in teaching contexts. | Research will be encouraged in different countries to know whether the model can be translated, adapted, and used in different context and cultures. |
| [62] | Grey et al., 2017 | *Advance Care Planning and Shared Decision-Making: An Interprofessional Role-Playing Workshop for Medical and Nursing Students* | 24 months | United States of America | (n = 85) medical and nursing students | Flipped classroom workshop | During the 2 h workshop, students complete four role-play ACP scenarios with the following roles: patient, family member, nurse, and physician. | Not reported | Survey | This role-play activity allows students to practice ACP and SDM, both with patient and family presence, and in premeeting rounds with the healthcare team. | The workshop can be utilized in many other levels of education. |
| [63] | Green and Levi, 2011 | *Teaching advance care planning to medical students with a computer-based decision aid* | Not reported | United States of America | (n = 133) medical students | Prospective, randomized controlled design | The multimedia decision aid helps prepare users to engage in advance care planning discussions by providing education material and exercises designed. | End-of-life decision making | Questionnaire | Use of a computer-based decision aid may be an effective way to teach medical students how to discuss advance care planning with cancer patients. | Look for a national study comparing this intervention with existing teaching modalities for advance care planning, and also invite other medical educators to examine the program. |
| [64] | Thompson and Stapley, 2011 | *Do educational interventions improve nurses' clinical decision-making and judgement? A systematic review* | Not reported | United Kingdom | Not reported | Systematic review | Studies published since 1960 reporting any educational intervention that aimed to improve nurses' clinical judgements or decision making were included. | Clinical or diagnostic reasoning | Not reported | Educational interventions to improve nurses' judgements and decisions are complex and the evidence from comparative studies does little to reduce the uncertainty about "what works". | Study design and reporting requires improvement to maximize the information contained in reports of educational interventions. |

**Table 1.** *Cont.*

| Ref No. | Author (s), Year of Publication | Title | Study Period | Country | Study Population and Sample Size (If Applicable) | Study Design | Methods/Methodology | SDM Definition | Data Collection Methods | Conclusion | Recommendations |
|---|---|---|---|---|---|---|---|---|---|---|---|
| [65] | Légaré et al., 2012 | *Training health professionals in shared decision-making: an international environmental scan* | Not reported | Canada | Not reported | Review | Environmental scan looking for programs that train health professionals in SDM | An interactive process in which patients and health professionals collaborate to choose healthcare. | Not reported | Health professional training programs in SDM vary widely in how and what they deliver, and evidence of their effectiveness is sparse. | The study suggests there is a need for international consensus on ways to address the variability in SDM training programs. |
| [66] | Légaré et al., 2012 | *Training family physicians in shared decision-making to reduce the overuse of antibiotics in acute respiratory infections: a cluster randomized trial* | 9 months | France | (n = 162) family physicians | Randomized trial | Twp study arms: DECISION+ 2 and control | Is recognized as an effective strategy for reducing the overuse of treatment options not clearly associated with benefits for all patients. | Questionnaire | The shared decision-making program DECISION+2 enhanced patient participation in decision making and led to fewer patients deciding to use antibiotics for acute respiratory infections. | Future studies should assess the effectiveness of SDM in other clinical areas. |
| [67] | Körner et al., 2013 | *Designing an interprofessional training programme for shared decision-making* | Not reported | Germany | (n = 36) patients and (n = 34) senior healthcare professionals | Cross-sectional mixed method | Two phases: focus groups of patients in the rehabilitation clinic and a second phase for the expert survey of senior healthcare professionals. | Is increasingly advocated as the ideal interaction model of external participation in patient–physician interaction. | Focus groups with patients and a survey of experts | The results of both assessments have been used to develop an interprofessional SDM training program for implementing internal and external participation in interprofessional teams in medical rehabilitation. | The approach ensures consideration of the important issues of internal and external participation and enhances acceptance of the implementation of training in these rehabilitation clinics. |
| [68] | Schell et al., 2013 | *Communication skills training for dialysis decision-making and end-of-life care in nephrology* | 1 month | United States of America | Not reported | Workshop design | NephroTalk was designed as a half-day workshop. | Helping patients define care goals, including end-of-life preferences. | Survey | NephroTalk is successful in improving preparedness among nephrology fellows for having difficult conversations about dialysis decision making and end-of-life care. | Disseminating NephroTalk to interested nephrology programs and encouraging education and awareness among nephrology educators. |
| [69] | Liaw et al., 2014 | *An interprofessional communication training using simulation to enhance safe care for a deteriorating patient* | Not reported | Singapore | (n = 127) medical and nursing students | Pre-test and post-test design | The program was conducted using full-scale simulation and communication strategies adapted from Team Strategies and Tools to Enhance Performance and Patient Safety (TeamSTEPPS). | Important factor in enhancing the students' confidence to communicate. | Questionnaire | The Sim-IPE has better prepared the medical and nursing students in communicating with one another in providing safe care for deteriorating patients. | Future studies could conduct a more rigorous research methodology such as randomized controlled trial. |
| [70] | Jo and An, 2015 | *Effects of an educational programme on shared decision-making among Korean nurses* | 1 month | Korea | (n = 41) nurses | Quasi-experimental study | Twenty nurses in the control group received no intervention, and twenty-one nurses in the experimental group received the educational programme on SDM. | Is a comprehensive concept of sharing information about treatment choices and decision methods based on the values and autonomy of the patients, families, doctors, and nurses. | Questionnaire | This study suggests that the educational programme on SDM was effective in increasing the moral sensitivity and attitude towards SDM among Korean nurses. | Future studies should investigate the effects of implementing similar programmers for longer periods. |
| [71] | Simmons et al., 2016 | *Shared decision-making in common chronic conditions: impact of a resident training workshop* | 4 months | United States of America | (n = 130) internal medicine and paediatric medicine residents | Curriculum development | Workshop curriculum for internal medicine residents to promote SDM in treatment decisions. | An interactive process that involves the clinician, the patient, and the best available clinical evidence to select the right medical test or treatment for each patient. | Written course evaluations and direct observation | Internal medicine residents had considerable gaps in SDM skills as measured in a baseline written exercise. | Additional studies are warranted to examine whether the workshop was successful in increasing residents' ability to implement skills in practice. |
| [72] | Légaré et al., 2011 | *Validating a conceptual model for an interprofessional approach to shared decision-making: a mixed methods study* | 3 months | Canada | (n = 79) stakeholders | Mixed Method | The participants were asked about the following: (1) propose changes to the IP-SDM model; (2) identify barriers and facilitators to the model's implementation in clinical practice; and (3) assess the model using a theory appraisal questionnaire. | An approach whereby practitioners and patients communicate around decisions, referring to the best available evidence and deliberating upon the consequences of each option. | Group interviews and individual interviews | Stakeholders validated the new IP-SDM model for primary care settings and proposed few modifications. | Future research should assess if the model helps implement SDM in IP clinical practice. |
| [73] | Halés and Hawryluck, 2008 | *An interactive educational workshop to improve end-of-life communication skills* | Not reported | Canada | (n = 6) members of varying disciplines | Pre-test and post-test design | A one-day interactive continuing education workshop. | A difficult and complex process as a result of differing perspectives among healthcare providers, patients, and families regarding ethics, benefits of treatment, culture, and religious beliefs. | Questionnaire | An interactive workshop can be a valuable educational intervention for building capacity and confidence in end-of-life communication skills and ethical and legal knowledge for HCPs. | Further research in this area should focus on evaluation of the lasting impact of this intervention on clinical practice. |

**Table 1.** *Cont.*

| Ref No. | Author (s), Year of Publication | Title | Study Period | Country | Study Population and Sample Size (If Applicable) | Study Design | Methods/Methodology | SDM Definition | Data Collection Methods | Conclusion | Recommendations |
|---|---|---|---|---|---|---|---|---|---|---|---|
| [74] | Wainwright et al., 2010 | *Novice and Experienced Physical Therapist Clinicians: A Comparison of How Reflection Is Used to Inform the Clinical Decision-Making Process* | Not reported | United States of America | (n = 3) clinicians | Qualitative research | Three participant pairs (each pair consisting of one novice and one experienced physical therapist). Case summaries of each participant provided the basis for within- and across-case analysis. | Reasoning that results in action. | Interview | The research provides information to educators, novice clinicians, and the clinicians who mentor these novices that may facilitate the development of mature clinical decision-making abilities. | The results of this study may be used by educators and employers to develop and structure learning experiences and mentoring opportunities to facilitate clinical decision-making abilities. |
| [75] | Keefe et al., 2002 | *Medical Students, Clinical Preventive Services, and Shared Decision-Making* | Not reported | United States of America | Not reported | Educational module | Not reported | Not reported | Videotaped discussion with a simulated patient | Medical students appear quite willing to accept SDM as a skill that they should have in working with patients, and this was the primary focus of the newly implemented module. | It would be helpful to provide students with more formative feedback and to develop faculty development programs around SDM. |
| [76] | Stephenson and Richardson, 2008 | *Building an Interprofessional Curriculum Framework for Health: A Paradigm for Health Function* | Not reported | United Kingdom | Not reported | Quasi-experimental | Adaption of ICF as a foundation for defining health status and for conceptualizing and formulating health-related client-focused problems. | Iterative process of reflection and reflexivity which takes into account wide evidence base relevant to the specific task of healthcare with the individual client and which can be developed in dialogue with other professionals. | Not reported | Client-focused practice and an iterative process of clinical reasoning based on a broad evidence base that conceptualizes healthcare as the maintenance, and promotion of health across the lifespan requires a re-conceptualizing of health. | The orientation of the curriculum needs to foster the development of collaboration and synergies of understanding between health professionals and between health professionals and clients of healthcare. |
| [77] | Edwards et al., 2005 | *Shared decision-making and risk communication in practice A qualitative study of GPs' experiences* | 4 months | United Kingdom | (n = 20) GPs | Qualitative study | The trial interventions comprised training in SDM skills and the use of risk communication materials. | Not reported | Exit interviews and questionnaire evaluations | The promotion of "patient involvement" appears likely to continue. | All the study findings require corroboration with a wider sample of practicing professionals. |
| [78] | Elwyn et al., 2005 | *Achieving involvement: process outcomes from a cluster randomized trial of shared decision-making skill development and use of risk communication aids in general practice* | Not reported | United Kingdom | (n = 352) patients and (n = 20) GPs | Cluster randomized design | Separate interventions to enhance clinician skills in either SDM or the use of risk communication aids were devised and piloted; they were provided to the clinicians before each active trial phase. | Process of involving patients in clinical decisions. | Questionnaires, audio taping, and patient interviews | The clinicians were able to acquire the skills to implement SDM competences and to use risk communication aids. | Progress towards greater patient involvement in healthcare decision making is possible, and skill development in this area should be incorporated into postgraduate professional development programmes. |
| [79] | Stacey et al., 2010 | *Shared decision-making models to inform an interprofessional perspective on decision-making: A theory analysis* | Not reported | Canada | Not reported | Theory analysis | Model of SDM; described concepts with relational statements. Two independently appraised models. | Not reported | Not reported | Most SDM models failed to encompass an interprofessional approach. Those that included at least two professionals met few of the elements of interprofessional collaboration and had limited description of SDM processes. | Appraisal of SDM models highlights the need for a model that is more inclusive of an interprofessional approach. |
| [80] | Curran, 2004 | *Interprofessional Education for Collaborative Patient-Centred Practice Research Synthesis Paper* | 13 months | Canada | Not reported | Research synthesis paper | Literature review and environmental scan undertaken by a multidisciplinary group of researchers. | Enables the separate and shared knowledge and skills of healthcare providers to synergistically influence the client/patient care provided. | Online survey and in-depth interviews | The purpose of this paper is to summarize the main themes emerging from the research report and discussion papers. | Readers are advised to consult the specific report or discussion paper for further elaboration and description. |

**Table 2.** Reported SDM interventions in included articles.

| Ref No. | Author (s), Year of Publication | Title | Disease/Medical Specialties | Settings/Clinical Area | Health Professionals' Involvement | Undergraduate or Postgraduate | Patient/Family Member Involvement | Type of Application | Educational Framework Learning Theory/ | Teaching Method/Activity/ Strategy/Delivery | Focuses on Knowledge, Attitudes, and/or Skills | Intervention Duration | SDM Model/SDM Tool/SDM Design | DM Components |
|---|---|---|---|---|---|---|---|---|---|---|---|---|---|---|
| [12] | Légaré et al., 2008 | *Advancing theories, models and measurement for an interprofessional approach to shared decision-making in primary care: a study protocol* | Chronic disease | Primary healthcare | Nurses and physicians | Not reported | Patients | Quality of patient decision | Not reported | Not reported | Skills and attitude | Not reported | Transactional and descriptive models | Essential elements and ideal elements |
| [13] | Müller et al., 2019 | *Strategies to evaluate healthcare provider trainings in shared decision-making (SDM): a systematic review of evaluation studies* | Not reported | Healthcare settings | Healthcare providers | Not reported | Not reported | Not reported | Not reported | Lectures, case studies, role play, and group discussion | Knowledge, skills and attitude | Not reported | Not reported | Not reported |
| [18] | Col et al., 2011 | *Interprofessional education about shared decision-making for patients in primary care settings* | Not reported | Primary healthcare | Not reported | Not reported | Patients and family members | Cross-cultural issues | Adult learning | Practical, interactive, and problem-based learning | Knowledge and skills | Not reported | Not reported | Not reported |
| [21] | Kryworuchko et al., 2013 | *Interventions for Shared Decision-Making About Life Support in the Intensive Care Unit: A Systematic Review* | End-of-life care | Intensive care unit | Healthcare team members | Not reported | Patients and family members | Intervention for end-of-life care | Not reported | Conference and brochure | Knowledge, skills, and attitudes | Not reported | SDM framework | 9 elements |
| [22] | Orchard et al., 2012 | *Assessment of Interprofessional Team Collaboration Scale (AITCS): Development and Testing of the Instrument* | Orthopaedic general surgery, acute mental health, and palliative care | Long-term care | Clinical psychologist, speech–language pathologist, nurse practitioner, child and youth worker, ward clerk, recreation therapist, therapy assistant, and orderly. | Undergraduate and postgraduate | Patients and family members | Collaboration in teams | Not reported | Not reported | Knowledge, skills, and attitudes | Not reported | Not reported | 19 items |
| [23] | Thomson et al., 2017 | *Making Decisions Better: an evaluation of an educational Intervention* | Not reported | Clinical settings | GP registrars and nurses | Undergraduate | Patients | Understanding of decision-making processes | Reflecting on learning | Interactive learning sessions | Skills | Not reported | Not reported | Not reported |
| [24] | Waldron et al., 2016 | *Development of a video-based education and process change intervention to improve advance cardiopulmonary resuscitation decision-making* | End-of-life care | Inpatient hospital | Junior doctors and consultants | Undergraduate | Patients and family members | Advance CPR decision making and communication | Adult educational theory | Education videos | Knowledge and skills | Not reported | CPR decision-making practices | (i) Knowing what to say; (ii) knowing how to say it; (iii) wanting to say it. |
| [25] | Sangaleti et al., 2017 | *Experiences and shared meaning of teamwork and interprofessional collaboration among health care professionals in primary health care settings: a systematic review* | Integrative medicine, family medicine | Primary healthcare | Not reported | Not reported | Not reported | Not reported | Not reported | Not reported | Not reported | Not reported | Not reported | Not reported |
| [26] | Nguyen et al., 2019 | *Conventional and Complementary Medicine Health Care Practitioners' Perspectives on Interprofessional Communication: A Qualitative Rapid Review* | Traditional and complementary medicine | Primary healthcare | Medical doctors, nurses, pharmacists, and other HCPs such as allied HCPs | Undergraduate and postgraduate | Patients and family members | Patient satisfaction, health literacy, treatment compliance, and quality of life | Not reported | Not reported | Knowledge, skills, and attitudes | Not reported | Not reported | Not reported |
| [27] | Shiao et al., 2019 | *Creation of nurse-specific integrated interprofessional collaboration and team efficiency scenario/video improves trainees' attitudes and performances* | Internal medicine | Simulation | Nurses, medical students, and other professions | Undergraduate | Simulated patients | Team efficiency | Experiential learning theory | Role play, videos, and discussion | Knowledge and skills | 4 weeks | Not reported | Not reported |

**Table 2.** *Cont.*

| Ref No. | Author (s), Year of Publication | Title | Disease/Medical Specialties | Settings/Clinical Area | Health Professionals' Involvement | Undergraduate or Postgraduate | Patient/Family Member Involvement | Type of Application | Educational Framework Learning Theory/ | Teaching Method/Activity/ Strategy/Delivery | Focuses on Knowledge, Attitudes, and/or Skills | Intervention Duration | SDM Model/SDM Tool/SDM Design | DM Components |
|---|---|---|---|---|---|---|---|---|---|---|---|---|---|---|
| [28] | Voogdt-Pruis et al., 2019 | *Improvement of shared decision-making in integrated stroke care: a before and after evaluation using a questionnaire survey* | Stroke | Outpatient rehabilitation and primary healthcare | Rehabilitation nurse, occupational therapist, physiotherapist, speech therapist, psychologist, rehabilitation specialist, and care manager | Postgraduate | Patients and family members | Stroke care | Not reported | Role play | Knowledge and skills | 1 year | Not reported | Not reported |
| [29] | Légaré et al., 2011 | *Interprofessionalism and shared decision-making in primary care:a stepwise approach towards a new model* | Not reported | Primary healthcare | Nurses, physicians, dietician, psychologist, anthropologist, and community health specialist | Undergraduate and postgraduate | Patients and family members | Patient's choices | Not reported | Workshop, presentations, and group discussion | Knowledge, skills, and attitudes | Not reported | IP-SDM model | 3 levels (micro, meso, macro) |
| [30] | McLaughlin et al., 2014 | *Rational and Experiential Decision-Making Preferences of Third-Year Student Pharmacists* | Not reported | University | Pharmacist students | Undergraduate | Not reported | Direct patient care and mitigation of medication errors | Not reported | Experiential decision-making activities | Knowledge and skills | Not reported | Not reported | Not reported |
| [31] | Chung et al., 2016 | *Educational interventions to train healthcare professionals in end-of-life communication: a systematic review and meta-analysis* | Palliative care | Not reported | Medical and nursing students | Undergraduate and postgraduate | Patients and family members | End-of-life communication | Not reported | Didactic lectures, small group discussions, role-play, direct observation, and feedback | Knowledge, skills, and attitudes | Not reported | Not reported | Not reported |
| [32] | Diouf et al., 2016 | *Training health professionals in shared decision-making: Update of an international environmental scan* | Generic, cancer, other chronic diseases | Primary healthcare | Physicians/residents, multiple professionals, and nurses | Not reported | Patients | Not reported | Not reported | Online course and traditional course | Knowledge, skills, and attitudes | Not reported | Not reported | Not reported |
| [33] | Johnsen et al., 2016 | *Teaching clinical reasoning and decision-making skills to nursing students: Design, development, and usability evaluation of a serious game* | Chronic obstructive pulmonary disease. | Home healthcare | Nursing students | Undergraduate | Simulated Patients | Clinical reasoning and decision-making skills | Clinical decision-making model and Bloom's taxonomy | Simulation technology | Skills | Not reported | TURF (Task, User, Representation, and Function) | Not reported |
| [34] | Kryworuchko et al., 2016 | *Factors influencing communication and decision-making about life-sustaining technology during serious illness: a qualitative study* | End-of-life care | Hospital | Staff physicians, residents, and nurses | Postgraduate | Patients and family members | Use of life-sustaining technology | Not reported | Not reported | Skills and attitudes | 47 min | Not reported | Not reported |
| [35] | Lestari et al., 2016 | *Understanding students' readiness for interprofessional learning in an Asian context: a mixed-methods study* | Not reported | University | Medical, nursing, midwifery, and dentistry students | Undergraduate | Simulated patients | Collaborative role | Not reported | Lectures | Knowledge, skills, and attitudes | Not reported | Not reported | Not reported |
| [36] | Lütgendorf-Caucig et al., 2017 | *Vienna Summer School on Oncology: how to teach clinical decision-making in a multidisciplinary environment* | Oncology | Hospital | Undergraduate medical students | Undergraduate | Not reported | Clinical decision-making in oncology | Kahneman model | Pre-module, presentations, classical lectures, workshops, and blended learning | Knowledge | 7 days | Not reported | Not reported |
| [37] | Légaré et al., 2018 | *Interventions for increasing the use of shared decision-making by healthcare professionals (Review)* | Cancer, cardiovascular diseases, psychiatric conditions | Primary and specialized care | Healthcare professionals (e.g., physicians, nurses, pharmacists, social workers) | Postgraduate | Patients and simulated patients | Not reported | Not reported | Not reported | Knowledge and skills | Not reported | Not reported | Not reported |
| [38] | Diendéré et al., 2019 | *How often do both core competencies of shared decision-making occur in family medicine teaching clinics?* | Family medicine | University teaching clinics | Family physicians, residents, nurses, and allied health professionals | Postgraduate | Patients | Chronic conditions, preventive care, and lifestyle issues | Not reported | Not reported | Skills | 4 to 6 days | Not reported | Not reported |

**Table 2.** *Cont.*

| Ref No. | Author (s), Year of Publication | Title | Disease/Medical Specialties | Settings/Clinical Area | Health Professionals' Involvement | Undergraduate or Postgraduate | Patient/Family Member Involvement | Type of Application | Educational Framework Learning Theory/ | Teaching Method/Activity/ Strategy/Delivery | Focuses on Knowledge, Attitudes, and/or Skills | Intervention Duration | SDM Model/SDM Tool/SDM Design | DM Components |
|---|---|---|---|---|---|---|---|---|---|---|---|---|---|---|
| [39] | Noguera et al., 2019 | *Student's Inventory of Professionalism (SIP): A Tool to Assess Attitudes towards Professional Development Based on Palliative Care Undergraduate Education* | Palliative care | University | Medical students | Undergraduate | Patients | Not reported | Not reported | Workshop | Knowledge and attitudes | Not reported | Wilkinson's framework | Not reported |
| [40] | Rajendran et al., 2019 | *Shared decision-making by United Kingdom osteopathic students: an observational study using the OPTION-12 Instrument* | Osteopathic | Teaching clinics | Fourth- and third-year students in the Osteopathic Educational Institute | Undergraduate | Patients | Long-term care management | Not reported | Not reported | Knowledge and skills | 7-week period | Not reported | Not reported |
| [41] | Allaire et al., 2012 | *What Motivates Family Physicians to Participate in Training Programs in Shared Decision-Making?* | Acute respiratory tract infections | Primary healthcare | Family physicians | Postgraduate | Patients | Level of agreement between the patient and the providers | Not reported | Workshops, videos, reflective exercises, and group discussion | Knowledge and skills | Workshops of 3 h each, for a total of 9 h over 4–6 months | DECISION+ | Major and minor components |
| [42] | Beitinger et al., 2014 | *Trends and perspectives of shared decision-making in schizophrenia and related disorders* | Mental Health | Clinics | Healthcare providers | Postgraduate | Patients and caregivers | Physicians' communication skills | Not reported | Not reported | Skills | Not reported | Decision aids | Not reported |
| [43] | Allen et al., 2020 | *Implementing a shared decision-making and cognitive strategy-based intervention: Knowledge user perspectives and recommendations* | Stroke | Rehabilitation hospitals | Occupational therapists, physical therapists, and speech language pathologists | Postgraduate | Patients | Knowledge and capacity among interprofessional team member and outcomes for patients discharged from inpatient stroke rehabilitation | Constructivist learning theory | Workshops | Knowledge and skills | 4 months | Not reported | Not reported |
| [44] | Kienlin et al., 2020 | *Ready for shared decision-making: Pretesting a training module for health professionals on sharing decisions with their patients* | Not reported | University/ college and hospital | Nurses, physicians, and health professional students | Undergraduate and postgraduate | Patients | Apply SDM in clinical practice | Not reported | Lecture | Knowledge and skills | 1 h vs. 2 h | Ready for SDM | Not reported |
| [45] | Keshmiri et al., 2020 | *The effect of interprofessional education on healthcare providers' intentions to engage in interprofessional shared decision-making: Perspectives from the theory of planned behaviour* | Emergency medicine | University hospitals | ED residents and nurses | Postgraduate | Patients | Communication, teamwork, and recognizing the roles of team members | Not reported | Case-based learning sessions | Skills and attitudes | Not reported | IP-SDM model | Not reported |
| [46] | Reed et al., 2017 | *Linking Essential Learning Outcomes and Interprofessional Collaborative Practice Competency in Health Science Undergraduates* | Not reported | University | Health profession students | Not reported | Patients | Perform skills and express emotional responses | Not reported | Situated activities | Skills and attitudes | Not reported | Not reported | Not reported |
| [47] | Wainwright et al., 2011 | *Factors That Influence the Clinical Decision-Making of Novice and Experienced Physical Therapists* | Cerebrovascular accident | Rehabilitation settings | Three clinician pairs, consisting of one novice and one experienced physical therapist | Undergraduate and postgraduate | Patients | Reasoning skills | Reflection in Clinical Decision-Making Revised Model | Observation and interview | Knowledge, skills and attitudes | Not reported | Schön's model | Informative factors and directive factors |
| [48] | Hansen et al., 2012 | *Life-Sustaining Treatment Decisions in the ICU for Patients with ESLD: A Prospective Investigation* | End-stage liver disease | Intensive care unit | Physicians, nurses, respiratory therapists, social workers, gastroenterology technician, and chaplain | Undergraduate and postgraduate | Patients and family members | Comfort care decisions | Not reported | Observation | Knowledge | 4–10 h each day, 3–6 morning hours and 1–4 h | Not reported | Not reported |

**Table 2.** *Cont.*

| Ref No. | Author (s), Year of Publication | Title | Disease/Medical Specialties | Settings/Clinical Area | Health Professionals' Involvement | Undergraduate or Postgraduate | Patient/Family Member Involvement | Type of Application | Educational Framework Learning Theory/ | Teaching Method/Activity/ Strategy/Delivery | Focuses on Knowledge, Attitudes, and/or Skills | Intervention Duration | SDM Model/SDM Tool/SDM Design | DM Components |
|---|---|---|---|---|---|---|---|---|---|---|---|---|---|---|
| [49] | Thompson et al., 2013 | *An agenda for clinical decision-making and judgement in nursing research and education* | Not reported | Not reported | Nurses | Not reported | Not reported | Nurse's decision making | Not reported | Not reported | Knowledge and skills | Not reported | Computerized decision support systems | Not reported |
| [50] | Giguere et al., 2012 | *Development of PRIDe: A tool to assess physicians' preference of role in clinical decision-making* | Acute respiratory infections | Not reported | Family physicians | Postgraduate | Patients | Decisional comfort | Not reported | Workshops, videos, reflective exercises, and group discussion | Knowledge, skills and attitudes | Not reported | Not reported | Not reported |
| [51] | Körner et al., 2012 | *Interprofessional SDM train-the-trainer programme "Fit for SDM": provider satisfaction and impact on participation* | Not reported | Medical rehabilitation clinic | Not reported | Postgraduate | Not reported | Not reported | Not reported | Not reported | Knowledge | Not reported | Not reported | Not reported |
| [52] | Sheridan et al., 2012 | *Shared decision-making for prostate cancer screening: the results of a combined analysis of 2 practice-based randomized controlled trials* | Prostate cancer | Not reported | Physicians | Postgraduate | Patients | Patients' participation | Not reported | Discussion and videos | Knowledge and attitudes | 1 h | O'Connor's Decisional Conflict Scale | 53 items |
| [53] | Yu et al., 2015 | *Impact of an interprofessional shared decision-making and goal-setting decision aid for patients with diabetes on decisional conflict—study protocol for a randomized controlled trial* | Diabetes | Primary healthcare | Physicians, nurses, dietitians, and pharmacists | Postgraduate | Patients and family members | Decisional conflict, diabetes distress | Knowledge-to-Action Framework | Training videos and patient education pamphlet | Knowledge, skills | Not reported | IP-SDM framework | 7 steps |
| [54] | Giguère et al., 2018 | *Tailoring and evaluating an intervention to improve shared decision-making among seniors with dementia, their caregivers, and healthcare providers: study protocol for a randomized controlled trial* | Dementia | Medicine unit | Physicians and residents; nurses and other health or social services professionals | Postgraduate | Patients and caregivers | Patient involvement, decisional comfort, patient quality of life, caregiver burden, and decisional regret | Not reported | e-learning | Attitudes | Not reported | Not reported | Not reported |
| [55] | Hendricks-Ferguson et al., 2018 | *Undergraduate students' perspectives of healthcare professionals' use of shared decision-making skills* | Not reported | University | Medical students | Undergraduate | Not reported | SDM responses | Not reported | Discussion | Knowledge | Not reported | Not reported | Not reported |
| [56] | Arenth et al., 2019 | *Teaching the Skill of Shared Decision-Making Utilizing a Novel Online Curriculum: a Blinded Randomized Controlled Pilot Study (S803)* | Palliative care | Children's hospital | Not reported | Postgraduate | Family members | Comfort care | Not reported | Video recorded in a simulated patient | Skills | Not reported | Not reported | Not reported |
| [57] | Hagoel et al., 2011 | *Interprofessional education about decision support for patients across cultures* | Not reported | Not reported | Not reported | Not reported | Patients | Cross-cultural issues | Adult learning | Scenarios, role playing, and videos | Knowledge, skills, and attitudes | Not reported | Explanatory models of illness or decision making | Not reported |
| [58] | Lown et al., 2011 | *Continuing professional development for interprofessional teams supporting patients in healthcare decision-making* | Not reported | University | Healthcare professionals | Undergraduate and postgraduate | Patients and family members | Decision support during the process of shared decision making | Six-step approach to curriculum development by Kern | Lectures, web-based targeted readings and other audiovisual resources, large and small group discussion, and problem-based learning | Knowledge, skills, and attitudes | Not reported | Not reported | 6 steps |

**Table 2.** *Cont.*

| Ref No. | Author (s), Year of Publication | Title | Disease/Medical Specialties | Settings/Clinical Area | Health Professionals' Involvement | Undergraduate or Postgraduate | Patient/Family Member Involvement | Type of Application | Educational Framework Learning Theory/ | Teaching Method/Activity/ Strategy/Delivery | Focuses on Knowledge, Attitudes, and/or Skills | Intervention Duration | SDM Model/SDM Tool/SDM Design | DM Components |
|---|---|---|---|---|---|---|---|---|---|---|---|---|---|---|
| [60] | Thistlethwaite et al., 2016 | *Introducing the individual Teamwork Observation and Feedback Tool (iTOFT): Development and description of a new interprofessional teamwork measure* | Not reported | University | Not reported | Not reported | Not reported | Not reported | Not reported | Not reported | Not reported | Not reported | Not reported | Not reported |
| [61] | Elwyn et al., 2017 | *A three-talk model for shared decision-making: multistage consultation process* | Internal medicine, family medicine, paediatrics | Not reported | Internal medicine, family medicine, and paediatric physicians | Postgraduate | Not reported | Patient's choices | Not reported | Web-based cases and simulations | Skills and attitudes | 12 months | Three-talk model | Not reported |
| [62] | Grey et al., 2017 | *Advance Care Planning and Shared Decision-Making: An Interprofessional Role-Playing Workshop for Medical and Nursing Students* | Nephrology | University | Medical students and undergraduate nursing students | Undergraduate | Patients and family members | Quality conversations between the provider and the patient | Not reported | Role-playing workshop | Knowledge, skills, and attitudes | 135 min flipped classroom for 2 years | Not reported | Not reported |
| [63] | Green and Levi, 2011 | *Teaching advance care planning to medical students with a computer-based decision aid* | Cancer, amyotrophic lateral sclerosis | University | Medical students | Undergraduate | Patients | Advance care planning and directive | Not reported | Question–answer format, clinical vignettes, video clips, lectures, and small group discussion | Knowledge and skills | Not reported | Not reported | Not reported |
| [64] | Thompson and Stapley, 2011 | *Do educational interventions improve nurses' clinical decision-making and judgement? A systematic review* | Not reported | Not reported | Not reported | Undergraduate and postgraduate | Patients | Decisional conflict | Social cognitive learning theory, decision analysis, and cognitive moral development theory | Critical thinking and problem-based learning | Skills | Not reported | The Outcome Present State model | Not reported |
| [65] | Légaré et al., 2012 | *Training health professionals in shared decision-making: an international environmental scan* | Palliative care, cardiovascular disease, prenatal screening, chronic pain, paediatrics, urology | Not reported | Any healthcare professions | Undergraduate and postgraduate | Patients and family members | Patient outcomes and organizational level | Not reported | Case-based discussion, small group educational session, role play, printed educational material, and feedback | Knowledge, skills and attitudes | Not reported | Ottawa Decision Support Framework | Not reported |
| [66] | Légaré et al., 2012 | *Training family physicians in shared decision-making to reduce the overuse of antibiotics in acute respiratory infections: a cluster randomized trial* | Acute respiratory infections | Practice teaching units | All family physicians, including physician teachers and residents | Postgraduate | Patients and family members | Decision to take antibiotics | Not reported | Online tutorial and workshop | Knowledge and attitudes | 2 h online tutorial followed by a 2 h interactive seminar | DECISION+2 | Not reported |
| [67] | Körner et al., 2013 | *Designing an interprofessional training programme for shared decision-making* | Not reported | Rehabilitation clinics | Medicine, psychotherapy, physical therapy, and nursing | Postgraduate | Patients | Management of feedback, talking with difficult team members, and moderate conflict discussion | Not reported | Focus group | Knowledge, skills, attitudes | Not reported | Model of integrated patient-centeredness and expanded model of SDM | Not reported |
| [68] | Schell et al., 2013 | *Communication skills training for dialysis decision-making and end-of-life care in nephrology* | Nephrology | University | Nephrology fellows | Postgraduate | Patients and family members | Delivering bad news and helping patients define care goals | The OncoTalk teaching model | Workshops | Knowledge and skills | 4 h workshop | NephroTalk | Specific skills demonstration and fellows' skills practice |
| [69] | Liaw et al., 2014 | *An interprofessional communication training using simulation to enhance safe care for a deteriorating patient* | End-of-life care | Simulation | Medical and nursing students | Undergraduate | Not reported | Communication skills between medical and nursing students | Presage–process–product (3P) model | Simulation and small group interprofessional learning | Skills | 3 h small group inter-professional learning | Not reported | Not reported |

**Table 2.** *Cont.*

| Ref No. | Author (s), Year of Publication | Title | Disease/Medical Specialties | Settings/Clinical Area | Health Professionals' Involvement | Undergraduate or Postgraduate | Patient/Family Member Involvement | Type of Application | Educational Framework Learning Theory/ | Teaching Method/Activity/ Strategy/Delivery | Focuses on Knowledge, Attitudes, and/or Skills | Intervention Duration | SDM Model/SDM Tool/SDM Design | DM Components |
|---|---|---|---|---|---|---|---|---|---|---|---|---|---|---|
| [70] | Jo and An, 2015 | *Effects of an educational programme on shared decision-making among Korean nurses* | End-of-life care | University hospitals | Nurses | Postgraduate | Patients and family members | End-of-life care performance, moral sensitivity, and attitude towards shared decision | Not reported | Education programmer | Knowledge and attitudes | 4 weeks | Not reported | Not reported |
| [71] | Simmons et al., 2016 | *Shared decision-making in common chronic conditions: impact of a resident training workshop* | Diabetes, depression, hypertension, and hyperlipidaemia | Clinics | Internal medicine residents | Postgraduate | Patients | Practice in shared decision-making skills | Not reported | Written case exercise, a short didactic presentation, and role-playing exercises | Skills | 1 h for PGY-1 residents and 2 h for PGY 2–4 residents | 6 Steps to Shared Decision-Making framework | 6 steps |
| [72] | Légaré et al., 2011 | *Validating a conceptual model for an interprofessional approach to shared decision-making: a mixed methods study* | Down syndrome | Primary healthcare | Health professionals, medical education, and the healthcare policy environment clinicians from primary healthcare teams | Not reported | Patients | Making a decision regarding prenatal screening for Down syndrome | Not reported | Short video illustrating an IP-SDM approach | Knowledge, skills, and attitudes | Not reported | Revised IP-SDM model | Various phases |
| [73] | Hales and Hawryluck, 2008 | *An interactive educational workshop to improve end-of-life communication skills* | End-of-life care | Hospital | Critical care providers of varying disciplines | Undergraduate | Patients and family members | Delivery of sensitive news | Experiential learning | Interactive workshops | Knowledge and skills | 45 min stations | Not reported | Not reported |
| [74] | Wainwright et al., 2010 | *Novice and Experienced Physical Therapist Clinicians: A Comparison of How Reflection Is Used to Inform the Clinical Decision-Making Process* | Cerebrovascular accident | Clinics | Three clinician pairs, consisting of one novice and one experienced physical therapist | Undergraduate and postgraduate | Patients | Reasoning skills | Reflection in Clinical Decision-Making Revised Model | Observation and interview | Knowledge and skills | Not reported | Schön's model | Attributes and behaviours of the participants |
| [75] | Keefe et al., 2002 | *Medical Students, Clinical Preventive Services, and Shared Decision-Making* | Cardiovascular disease and cancer | Simulation | Medical students | Undergraduate | Patients | Screening cancer and lipid profile | Model adapted from Braddock and colleagues | Not reported | Knowledge and skills | Not reported | Not reported | Not reported |
| [76] | Stephenson and Richardson, 2008 | *Building an Interprofessional Curriculum Framework for Health: A Paradigm for Health Function* | Chronic disease | University | Physicians, nurses, and occupational therapists | Undergraduate | Family members | Ethical decision | Not reported | Case study | Attitudes and knowledge | 3 of 5 sections taught in a course semester | Not reported | Not reported |
| [77] | Edwards et al., 2005 | *Shared decision-making and risk communication in practice A qualitative study of GPs' experiences* | Surgery | Health authority | General practitioners (GPs) | Postgraduate | Patients | Patient involvement | Work-based experiential learning | Workshops | Skills | Not reported | Not reported | Not reported |
| [78] | Elwyn et al., 2005 | *Achieving involvement: process outcomes from a cluster randomized trial of shared decision-making skill development and use of risk communication aids in general practice* | Patients with known atrial fibrillation, prostatitis, menorrhagia, or menopausal symptoms | Urban and rural general practices | Recently qualified GPs | Postgraduate | Patients | Risk communication | Not reported | Workshops | Skills | Not reported | Simple risk communication aids | Not reported |
| [79] | Stacey et al., 2010 | *Shared decision-making models to inform an interprofessional perspective on decision-making: A theory analysis* | Not reported | Not reported | Not reported | Not reported | Not reported | Not reported | Not reported | Not reported | Knowledge and skills | Not reported | Not reported | Not reported |
| [80] | Curran, 2004 | *Interprofessional Education for Collaborative Patient-Centred Practice Research Synthesis Paper* | Not reported | Not reported | Not reported | Undergraduate | Patients and family members | Patient and provider satisfaction, patient outcomes | Experiential learning strategy and adult learning theory | Cooperative learning, small group learning, case-based learning, and problem-based learning | Knowledge, skills, and attitudes | Not reported | IECPCP Synthesis Framework | Separate components within the framework |

**Table 3.** Reported outcomes in included articles.

| Ref No. | Author (s), Year of Publication | Title | Evaluation Framework | Type of Outcome | SDM Measures/Instruments | Summative and/or Formative Assessment | Results |
|---------|---------------------------------|-------|----------------------|-----------------|--------------------------|---------------------------------------|---------|
| [12] | Légaré et al., 2008 | *Advancing theories, models and measurement for an interprofessional approach to shared decision-making in primary care: a study protocol* | Evaluation by McDowell and Newell and by Tremblay and collaborators | Impact on health systems and organizations | Measurement tools for enhancing an interprofessional approach to SDM in primary healthcare | Not reported | The authors of this systematic review concluded that it was important to study communication and decision making in relatively mundane contexts such as suggesting that SDM in primary healthcare contexts had been satisfactorily addressed. |
| [13] | Müller et al., 2019 | *Strategies to evaluate healthcare provider trainings in shared decision-making (SDM): a systematic review of evaluation studies* | Kirkpatrick's evaluation levels and Quadruple Aim framework | Students' professional development | Not reported | Summative and formative | Identified evaluation outcomes covered all categories of the proposed framework. |
| [18] | Col et al., 2011 | *Interprofessional education about shared decision-making for patients in primary care settings* | Not reported | Patient care | Patient decision aids | Not reported | A series of teaching methods using principles from adult learning. |
| [21] | Kryworuchko et al., 2013 | *Interventions for Shared Decision-Making About Life Support in the Intensive Care Unit: A Systematic Review* | Not reported | Patient's value and preferences | Not reported | Not reported | The interventions were not harmful; they decreased family member anxiety and distress, shortened intensive care unit stay, but did not affect patient mortality. |
| [22] | Orchard et al., 2012 | *Assessment of Interprofessional Team Collaboration Scale (AITCS): Development and Testing of the Instrument* | Not reported | Team collaboration | Assessment of Interprofessional Team Collaboration Scale (AITCS) | Not reported | The AITCS can help healthcare teams enhance their development as teams by focusing attention on areas their members view as not being collaborative. |
| [23] | Thomson et al., 2017 | *Making Decisions Better: an evaluation of an educational Intervention* | Not reported | Understanding of decision-making processes and application to clinical practice | Joint Practice—PRE and POST | Formative | Participation in the learning sessions significantly improved self-reported understanding of decision-making processes and application to clinical practice. The extended learning sessions did not provide additional benefits over and above 2 half days or 1 whole day learning sessions. |
| [24] | Waldron et al., 2016 | *Development of a video-based education and process change intervention to improve advance cardiopulmonary resuscitation decision-making* | Not reported | Patients' preferences | "Goals of Patient Care" (GOPC) form and Supportive and Palliative Care Indicators Tool (SPICT) tool | Not reported | CPR decision-making analysis: (i) knowing what to say; (ii) knowing how to say it; (iii) wanting to say it. |
| [25] | Sangaleti et al., 2017 | *Experiences and shared meaning of teamwork and interprofessional collaboration among health care professionals in primary health care settings: a systematic review* | Not reported | Team collaboration | Not reported | Not reported | Not reported |

**Table 3.** *Cont.*

| Ref No. | Author (s), Year of Publication | Title | Evaluation Framework | Type of Outcome | SDM Measures/Instruments | Summative and/or Formative Assessment | Results |
|---------|-------------------------------|-------|---------------------|-----------------|--------------------------|--------------------------------------|---------|
| [26] | Nguyen et al., 2019 | *Conventional and Complementary Medicine Health Care Practitioners' Perspectives on Interprofessional Communication: A Qualitative Rapid Review* | Not reported | Not reported | Not reported | Not reported | Four key themes were identified that impact IPC: medical dominance, clarity of HCP roles, a shared vision, and education and training. |
| [27] | Shiao et al., 2019 | *Creation of nurse-specific integrated interprofessional collaboration and team efficiency scenario/video improves trainees' attitudes and performances* | Kirkpatrick's Model | Team performance | Assessment of Interprofessional Team Collaboration Scale (AITCS) Attitudes Toward Interprofessional Health Care Teams Scale (ATHCTS) | Formative | Nursing trainees in intervention group gave high satisfaction score to this IIT intervention and increase in instructor-assessed team performance in the "partnership," "cooperation," and "shared decision making". |
| [28] | Voogdt-Pruis et al., 2019 | *Improvement of shared decision-making in integrated stroke care: a before and after evaluation using a questionnaire survey* | Not reported | Patients' preferences | Not reported | Formative | Healthcare professionals provided 8 recommendations for adoption of SDM in integrated stroke care. |
| [29] | Légaré et al., 2011 | *Interprofessionalism and shared decision-making in primary care: a stepwise approach towards a new model* | Not reported | Patients' value and preferences | Nine theory appraisal criteria | Not reported | The model has the potential to improve traditional decision-making processes and working practices currently exercised in many industrialized healthcare systems. |
| [30] | McLaughlin et al., 2014 | *Rational and Experiential Decision-Making Preferences of Third-Year Student Pharmacists* | Not reported | Clinical problem-solving skills | The Rational-Experiential Inventory (REI-40) | Not reported | All correlations between REI-40 scores and incoming grade point average (GPA) and Pharmacy College Admission Test (PCAT) scores were weak. |
| [31] | Chung et al., 2016 | *Educational interventions to train healthcare professionals in end-of-life communication: a systematic review and meta-analysis* | Kirkpatrick's Model | Students' self-efficacy, knowledge, improvements in communication | Not reported | Not reported | Twenty were studies of educational interventions and were reviewed in this paper. |
| [32] | Diouf et al., 2016 | *Training health professionals in shared decision-making: Update of an international environmental scan* | Not reported | Training satisfaction | Not reported | Not reported | A total of 94 new eligible programs in 4 new countries and 2 new languages, for a total of 148 programs produced from 1996 to 2015. |
| [33] | Johnsen et al., 2016 | *Teaching clinical reasoning and decision-making skills to nursing students: Design, development, and usability evaluation of a serious game* | Not reported | Not reported | Cognitive walkthrough evaluations | Not reported | The SG was perceived as being realistic, clinically relevant, and at an adequate level of complexity for the intended users. |
| [34] | Kryworuchko et al., 2016 | *Factors influencing communication and decision-making about life-sustaining technology during serious illness: a qualitative study* | Flanagan's critical incident technique | Healthcare professionals, patient and family engagement | DECIDE quantitative | Not reported | Several key factors that influenced communication and decision making about life-sustaining technology. |

**Table 3.** *Cont.*

| Ref No. | Author (s), Year of Publication | Title | Evaluation Framework | Type of Outcome | SDM Measures/Instruments | Summative and/or Formative Assessment | Results |
|---------|--------------------------------|-------|---------------------|-----------------|--------------------------|--------------------------------------|---------|
| [35] | Lestari et al., 2016 | *Understanding students' readiness for interprofessional learning in an Asian context: a mixed-methods study* | Not reported | Not reported | Readiness for Interprofessional Learning Scale (RIPLS) | Not reported | Medical students seemed to be the most prepared for IPE. |
| [36] | Lütgendorf-Caucig et al., 2017 | *Vienna Summer School on Oncology: how to teach clinical decision-making in a multidisciplinary environment* | Not reported | Students' knowledge acquisition | Compulsory pre-VSSO and post-VSSO single choice questionnaire | Formative | Most students' comments about the VSSO were very positive. |
| [37] | Légaré et al., 2018 | *Interventions for increasing the use of shared decision-making by healthcare professionals (Review)* | Not reported | Primary and secondary outcomes | Not reported | Not reported | There was insufficient information to determine the effects on decision regret, physical- or mental-health-related quality of life, or consultation length or costs. |
| [38] | Diendéré et al., 2019 | *How often do both core competencies of shared decision-making occur in family medicine teaching clinics?* | Not reported | Patients' values clarification | The OPTION 5 (observing patient involvement in decision-making) | Formative | The core elements of SDM occurred together in nearly two-thirds of visits without any active intervention. |
| [39] | Noguera et al., 2019 | *Student's Inventory of Professionalism (SIP): A Tool to Assess Attitudes towards Professional Development Based on Palliative Care Undergraduate Education* | Not reported | Students' performance in educational activities | Student's Inventory of Professionalism (SIP) | Not reported | "Student's Inventory on Professionalism" to indicate with the name the construct explored and that it is grounded in students' perceptions. |
| [40] | Rajendran et al., 2019 | *Shared decision-making by United Kingdom osteopathic students: an observational study using the OPTION-12 Instrument* | Not reported | SDM behaviours | Validated OPTION-12 (O12) instrument (observing patient involvement) scale | Formative | No significant differences between O12 score of the third- and fourth-year students, which implies that the extra year of clinical teaching and supervision does not result in a higher engagement of SDM within the undergraduate teaching clinic. |
| [41] | Allaire et al., 2012 | *What Motivates Family Physicians to Participate in Training Programs in Shared Decision-Making?* | Not reported | Decision conflict level | DECISION+ and decision conflict scale | Not reported | CPD developers should promote their programs as interesting, enjoyable, and professionally stimulating. |
| [42] | Beitinger et al., 2014 | *Trends and perspectives of shared decision-making in schizophrenia and related disorders* | Not reported | Patients' self-advocacy | 9-item SDM Questionnaire (SDM-Q-9), SDM scale sum score, sum score of the decision-making subscale of the API, physician ratings of patient behaviour, OPTION scale, 28-item Empowerment Scale, adapted version of "Elements of Informed Decision-Making Scale", COMRADE, patient rated | Not reported | There are only a few interventional studies measuring the outcome of SDM; existing research constantly shows positive, but small effects. |

**Table 3.** *Cont.*

| Ref No. | Author (s), Year of Publication | Title | Evaluation Framework | Type of Outcome | SDM Measures/Instruments | Summative and/or Formative Assessment | Results |
|---|---|---|---|---|---|---|---|
| [43] | Allen et al., 2020 | *Implementing a shared decision-making and cognitive strategy-based intervention: Knowledge user perspectives and recommendations* | Integrated promoting action on research implementation in health services (iPARIHS) framework | Enhanced knowledge and capacity among interprofessional team members | Cognitive Orientation to daily Occupational Performance (CO-OP) | Not reported | Participants suggested there needs to be specific training and a familiarity with the language across professions and among patients to ensure consistency in documentation, verbal communication, and person-centred care. |
| [44] | Kienlin et al., 2020 | *Ready for shared decision-making: Pretesting a training module for health professionals on sharing decisions with their patients* | The Medical Research Council Complex Interventions Framework, Kirkpatrick's model | Improve communication and patient involvement | Not reported | Summative and formative | Participants gained knowledge of SDM relevant for improved communication. This study has only evaluated the first two levels of the Kirkpatrick's model, but the intention is to make changes based on these findings and evaluate the other levels involvement. |
| [45] | Keshmiri et al., 2020 | *The effect of interprofessional education on healthcare providers' intentions to engage in interprofessional shared decision-making: Perspectives from the theory of planned behaviour* | Not reported | Team collaboration | TPB-based questionnaire | Not reported | The qualitative data analysis showed two main categories of "team-based facilitators" and "contextual challenges" as the main affecting factors in the engagement of participant in IP-SDM. |
| [46] | Reed et al., 2017 | *Linking Essential Learning Outcomes and Interprofessional Collaborative Practice Competency in Health Science Undergraduates* | Not reported | Students' ethical reasoning decision | Interprofessional Collaborative Practice (IPCEP) Core Competency of Values/Ethics | Not reported | Most students demonstrated adequate achievement of the Interprofessional Collaborative Practice (IPCEP) Core Competency of Values/Ethics. |
| [47] | Wainwright et al., 2011 | *Factors That Influence the Clinical Decision-Making of Novice and Experienced Physical Therapists* | Reflection-on-action (ROA) | Clinical decision-making abilities | Semi-Structured Interview Question Guide: Think-Aloud Videotape Analysis Interviews | Not reported | The factors that influenced clinical decision making were categorized as informative or directive. Novice participants relied more on informative factors, whereas experienced participants were more likely to rely on directive factors. |
| [48] | Hansen et al., 2012 | *Life-Sustaining Treatment Decisions in the ICU for Patients with ESLD: A Prospective Investigation* | Signal detection theory, judgement analysis | Comfort care decisions | Not reported | Not reported | Findings suggest that including patients and family members in non-immediate lifesaving decisions and verifying early their understanding may help to improve the decision-making process. |
| [49] | Thompson et al., 2013 | *An agenda for clinical decision-making and judgement in nursing research and education* | Not reported | Improve quality in healthcare systems | Not reported | Not reported | CDSS can help improve practice but is limited. |
| [50] | Gigue're et al., 2012 | *Development of PRIDe: A tool to assess physicians' preference of role in clinical decision-making* | Not reported | Health professionals' attitude towards SDM | Theory of Planned Behaviour-based questionnaire | Formative | Five items for potential inclusion in PRIDe. The results of these items were pooled, and their reliability and validity explored. |

**Table 3.** *Cont.*

| Ref No. | Author (s), Year of Publication | Title | Evaluation Framework | Type of Outcome | SDM Measures/Instruments | Summative and/or Formative Assessment | Results |
|---|---|---|---|---|---|---|---|
| [51] | Körner et al., 2012 | *Interprofessional SDM train-the-trainer programme "Fit for SDM": provider satisfaction and impact on participation* | Not reported | SDM skills and satisfaction | Not reported | Not reported | Not reported |
| [52] | Sheridan et al., 2012 | *Shared decision-making for prostate cancer screening: the results of a combined analysis of 2 practice-based randomized controlled trials* | "PSA is a Decision" | Patients' knowledge | 3-item uncertainty subscale from O'Connor's Decisional Conflict Scale | Not reported | Participants in the control group were additionally slightly less likely to consider prostate cancer screening a decision and slightly more likely to have key knowledge about prostate cancer screening. |
| [53] | Yu et al., 2015 | *Impact of an interprofessional shared decision-making and goal-setting decision aid for patients with diabetes on decisional conflict—study protocol for a randomized controlled trial* | Not reported | Improve clinical outcomes | Patient questionnaires of validated scales—SPIRIT checklist | Not reported | The development of an evidence-based SDM intervention for patients with diabetes and other conditions that was framed by the IP-SDM model and followed a user-centred approach. |
| [54] | Giguère et al., 2018 | *Tailoring and evaluating an intervention to improve shared decision-making among seniors with dementia, their caregivers, and healthcare providers: study protocol for a randomized controlled trial* | CollaboRATE instrument | Healthcare empowerment, caregiver burden, patient quality of life, and decisional regret | QoL-AD questionnaire | Not reported | Not reported |
| [55] | Hendricks-Ferguson et al., 2018 | *Undergraduate students' perspectives of healthcare professionals' use of shared decision-making skills* | Not reported | Understanding of SDM and ethical principles | Student reflection assignments | Not reported | Not reported |
| [56] | Arenth et al., 2019 | *Teaching the Skill of Shared Decision-Making Utilizing a Novel Online Curriculum: a Blinded Randomized Controlled Pilot Study (S803)* | Not reported | Comfort care decisions | Validated scoring tool for the degree of shared decision making | Not reported | Regression analysis demonstrated the odds of improved performance in mean total score for intervention groups was 39.78 times greater than that of the control group. |
| [57] | Hagoel et al., 2011 | *Interprofessional education about decision support for patients across cultures* | Not reported | Not reported | Not reported | Not reported | Not reported |
| [58] | Lown et al., 2011 | *Continuing professional development for interprofessional teams supporting patients in healthcare decision-making* | Kirkpatrick's Model | Interpersonal and communication skills | OPTION instrument COMRADE instrument Team Dimensions Rating Form Collaboration and Satisfaction About Care Decisions | Summative and formative | The study describes a model that can be used to design, implement, and evaluate continuing education curricula in IP-SDM and decision support. |

**Table 3.** *Cont.*

| Ref No. | Author (s), Year of Publication | Title | Evaluation Framework | Type of Outcome | SDM Measures/Instruments | Summative and/or Formative Assessment | Results |
|---|---|---|---|---|---|---|---|
| [59] | Neville et al., 2013 | *Team decision-making: design, implementation and evaluation of an interprofessional education activity for undergraduate health science students* | Not reported | Team effectiveness | Readiness for Interprofessional Learning Scale, Interdisciplinary Education Perception Scale, and the Role Perception Questionnaires | Not reported | Students were willing to share their knowledge and skills as a way of understanding clinical problems in the workplace and had professionally oriented perceptions and related affective domains. |
| [60] | Thistlethwaite et al., 2016 | *Introducing the individual Teamwork Observation and Feedback Tool (iTOFT): Development and description of a new interprofessional teamwork measure* | Not reported | Clinical teamwork experience | Individual Teamwork Observation and Feedback Tool (iTOFT) | Formative | Not reported |
| [61] | Elwyn et al., 2017 | *A three-talk model for shared decision-making: multistage consultation process* | SHARE (Seek participation, Help comparison, Assess values, Reach decision, Evaluate decision) | Patients' preferences | Not reported | Not reported | A new three-talk model of SDM is proposed, based on "team talk", "option talk", and "decision talk", to depict a process of collaboration and deliberation. |
| [62] | Grey et al., 2017 | *Advance Care Planning and Shared Decision-Making: An Interprofessional Role-Playing Workshop for Medical and Nursing Students* | Not reported | Teaching effectiveness | Not reported | Formative | Advance care planning (ACP) exposure during student training helps trainees recognize the impact of high-quality interprofessional conversations on the care patients want and ultimately receive. |
| [63] | Green and Levi, 2011 | *Teaching advance care planning to medical students with a computer-based decision aid* | Not reported | Students' knowledge, skill, and satisfaction | Pre-intervention and post-intervention evaluations and evaluation of student performance by patients, 17-item true/false and multiple-choice test, self-assessment instrument, 12-item instrument that addressed students' communication skills | Formative | Patients in the decision aid group were more satisfied with the advance care planning method and with several aspects of student performance. |
| [64] | Thompson and Stapley, 2011 | *Do educational interventions improve nurses' clinical decision-making and judgement? A systematic review* | Outcome Present State model | Patient outcomes | Not reported | Formative | From 5262 initial citations 24 studies were included in the review. The effectiveness and efficacy of interventions was mixed. |
| [65] | Légaré et al., 2012 | *Training health professionals in shared decision-making: an international environmental scan* | Kirkpatrick's Model | Patient outcomes and organizational level | Not reported | Not reported | A total of 54 programs conducted between 1996 and 2011 in 14 countries and 10 languages. |
| [66] | Légaré et al., 2012 | *Training family physicians in shared decision-making to reduce the overuse of antibiotics in acute respiratory infections: a cluster randomized trial* | Not reported | Patients' adherence to the decision | Decisional Conflict Scale | Not reported | The percentage of patients who decided to use antibiotics after consultation was 52.2% in the control group and 27.2% in the DECISION+2 group. |

Table 3. *Cont*.

| Ref No. | Author (s), Year of Publication | Title | Evaluation Framework | Type of Outcome | SDM Measures/Instruments | Summative and/or Formative Assessment | Results |
|---------|----------|-------|----------------------|-----------------|--------------------------|---------------------------------------|---------|
| [67] | Körner et al., 2013 | *Designing an interprofessional training programme for shared decision-making* | Not reported | External participation (interaction between patient and healthcare professionals) and internal participation (communication, coordination, and cooperation in the interprofessional team) | Not reported | Not reported | The results indicate the importance of internal and external participation in interprofessional settings. |
| [68] | Schell et al., 2013 | *Communication skills training for dialysis decision-making and end-of-life care in nephrology* | Not reported | End-of-life preferences | Not reported | Not reported | The results presented highlight the need for structured communication education in nephrology programs. |
| [69] | Liaw et al., 2014 | *An interprofessional communication training using simulation to enhance safe care for a deteriorating patient* | Not reported | Students' self-confidence | The C-scale with 10-point scales | Formative | Both medicine and nursing groups demonstrated a significant improvement on post-test score from pre-test score for self-confidence and perception. The participants were highly satisfied with their simulation learning. |
| [70] | Jo and An, 2015 | *Effects of an educational programme on shared decision-making among Korean nurses* | Not reported | End-of-life care performance, moral sensitivity, and attitude towards SDM | End-of-life care performance scale, Moral Sensitivity Questionnaire, attitude towards shared decision-making scale | Not reported | The experimental group showed significantly higher scores in moral sensitivity and attitude towards SDM after the intervention compared with the control group. |
| [71] | Simmons et al., 2016 | *Shared decision-making in common chronic conditions: impact of a resident training workshop* | Not reported | Health behaviours, adherence, health outcomes | Not reported | Formative | Residents were involved in the development of the workshop and helped identify key content, suggested framing for difficult topics, and confirmed the need for the skills workshop. |
| [72] | Légaré et al., 2011 | *Validating a conceptual model for an interprofessional approach to shared decision-making: a mixed methods study* | Not reported | Interprofessional collaboration | Theory appraisal questionnaire scale | Not reported | Stakeholders suggested placing the patient at its centre; extending the concept of family to include significant others; clarifying outcomes; highlighting the concept of time; merging the micro, meso, and macro levels in one figure. |
| [73] | Hales and Hawryluck, 2008 | *An interactive educational workshop to improve end-of-life communication skills* | Not reported | End of life communication, ethical and legal knowledge for clinicians | Preworkshop and postworkshop evaluations | Formative | High overall perception of success and achievement of educational objectives. |
| [74] | Wainwright et al., 2010 | *Novice and Experienced Physical Therapist Clinicians: A Comparison of How Reflection Is Used to Inform the Clinical Decision-Making Process* | Reflection on- action (ROA) | Clinical decision-making abilities | Semi-Structured Interview Question Guide: Think-Aloud Videotape Analysis Interviews | Formative | The data illustrate the theme of reflection as it is used to inform the clinical decision-making process. |

**Table 3.** *Cont.*

| Ref No. | Author (s), Year of Publication | Title | Evaluation Framework | Type of Outcome | SDM Measures/Instruments | Summative and/or Formative Assessment | Results |
|---------|--------------------------------|-------|---------------------|-----------------|--------------------------|--------------------------------------|---------|
| [75] | Keefe et al., 2002 | *Medical Students, Clinical Preventive Services, and Shared Decision-Making* | Not reported | Skills development | Checklist on the elements of SDM | Summative and formative | Explicit model that allows students to demonstrate a process for SDM is a good introductory tool. |
| [76] | Stephenson and Richardson, 2008 | *Building an Interprofessional Curriculum Framework for Health: A Paradigm for Health Function* | Not reported | Client function | Not reported | Not reported | The framework can promulgate a paradigm of practice within an interprofessional dialogue of healthcare. |
| [77] | Edwards et al., 2005 | *Shared decision-making and risk communication in practice A qualitative study of GPs' experiences* | Not reported | SDM skills | Not reported | Not reported | The GPs indicated positive attitudes towards involving patients and described positive effects on their consultations. |
| [78] | Elwyn et al., 2005 | *Achieving involvement: process outcomes from a cluster randomized trial of shared decision-making skill development and use of risk communication aids in general practice* | OPTION: observing patients, multilevel modelling involvement | SDM skills | OPTION scale | Formative | Clinicians increased the proportion of consultations in which they used several categories of risk information after the risk communication training intervention. |
| [79] | Stacey et al., 2010 | *Shared decision-making models to inform an interprofessional perspective on decision-making: A theory analysis* | Medical Research Council framework | Interprofessional collaboration | Not reported | Not reported | The 15 unique models included 18 core concepts. Of two models that included more than one health professional collaborating with the patient, one included 3 of 10 elements of interprofessional collaboration and the other included 1 element. |
| [80] | Curran, 2004 | *Interprofessional Education for Collaborative Patient-Centred Practice Research Synthesis Paper* | Not reported | Patient and provider satisfaction, patient outcomes | Team Oral Structured Clinical Examination or (TOSCE) | Not reported | Main factors determinants and elements as they relate to the micro, meso, and macro levels. |

## 2.5. Collecting, Summarizing, and Reporting the Data

Data synthesis was conducted according to the research questions. Data analysis involved quantitative frequency analysis and qualitative thematic analysis. Descriptive analyses, including proportions and means, were used to characterize identified studies and interventions. Summaries of extracted data are presented in text and tabular form (Table 4).

**Table 4.** Characteristics (peer-reviewed).

| A. Study Characteristics/General Information | |
|:---:|:---:|
| **Country** | |
| Canada | **19 (30%)** |
| USA | **16 (24%)** |
| UK | **9 (14%)** |
| Australia | **4 (7%)** |
| Germany | **4 (7%)** |
| Other * | **11 (18%)** |
| **Study design** | |
| Review | **14 (22%)** |
| Before and after evaluation study Pre-intervention and post-intervention | **5 (8%)** |
| Explanatory, qualitative study | **6 (10%)** |
| Instrument design, instrument validation, curriculum development, curriculum design | **15 (24%)** |
| Mixed-method design | **9 (14%)** |
| Cross-sectional design | **1 (2%)** |
| Randomized controlled trial | **9 (14%)** |
| Quasi-experimental, survey, action research | **1 (2%)** |
| N/A | **3 (4%)** |
| **B. SDM interventions** | |
| **Disease(s)/medical specialties** | |
| Down syndrome | **1 (2%)** |
| Family medicine/internal medicine/chronic diseases, including diabetes, stroke, liver diseases, lung diseases, cardiovascular diseases | **22 (34%)** |
| End-of-life/palliative care/oncology | **14 (21%)** |
| Orthopaedic/osteopathic/surgery | **3 (5%)** |
| Integrative medicine/traditional and complementary medicine | **2 (3%)** |
| Mental health | **1 (2%)** |
| Emergency medicine | **1 (2%)** |
| Not reported | **19 (31%)** |
| **Settings/clinical area** | |
| Primary healthcare | **9 (14%)** |
| Intensive care unit | **2 (4%)** |
| Long-term care/home healthcare | **2 (4%)** |
| Hospital | **10 (15%)** |
| Simulation | **3 (5%)** |

**Table 4.** *Cont.*

| | |
|---|---|
| Outpatient clinic | **7 (11%)** |
| University teaching clinic | **17 (26%)** |
| Health authority | **1 (2%)** |
| Urban and rural general practices | **1 (2%)** |
| Not reported | **11 (17%)** |
| **Undergraduate and/or postgraduate** | |
| Undergraduate | **18 (29%)** |
| Postgraduate | **22 (35%)** |
| Both | **11 (17%)** |
| Not reported | **12 (19%)** |
| **Patient/family member involvement** | |
| Patient | **28 (44%)** |
| Family member | **2 (4%)** |
| Both | **21 (32%)** |
| None | **12 (20%)** |
| **Type of decisions/applications** | |
| Decision quality | **6 (9%)** |
| Communication and collaboration | **18 (28%)** |
| Patient care, satisfaction | **10 (16%)** |
| Healthcare choice | **8 (13%)** |
| Application in clinical practice | **2 (3%)** |
| SDM processes | **4 (6%)** |
| Clinical reasoning | **3 (5%)** |
| Use of technology | **1 (2%)** |
| Ethical decision | **1 (2%)** |
| Cultural issue | **2 (3%)** |
| Not reported | **8 (13%)** |
| **Teaching method/activity/strategy/delivery** | |
| Video | **4 (6%)** |
| Role play | **4 (6%)** |
| Observation | **3 (5%)** |
| Interactive learning sessions, discussion | **9 (15%)** |
| Case-based learning | **5 (8%)** |
| Lectures | **5 (8%)** |
| Online course | **3 (5%)** |
| Blended learning | **1 (2%)** |
| Simulation | **3 (5%)** |
| Workshop | **11 (17%)** |
| Not reported | **15 (23%)** |
| **Focuses on knowledge, attitudes, skills** | |
| Knowledge | **4 (6%)** |
| Attitudes | **1 (2%)** |

**Table 4.** *Cont.*

| | |
|---|---|
| Skills | **10 (16%)** |
| All | **17 (27%)** |
| Knowledge and attitudes | **5 (8%)** |
| Knowledge and skills | **19 (30%)** |
| Attitudes and skills | **5 (8%)** |
| N/A | **2 (3%)** |
| **Intervention duration** | |
| Less than 2 h | **5 (8%)** |
| 3–4 h | **3 (5%)** |
| 1–7 days | **3 (5%)** |
| 1–8 weeks | **3 (5%)** |
| 2–12 months | **5 (8%)** |
| Longer than 12 months | **1 (2%)** |
| Not reported | **43 (67%)** |
| **C. Outcomes** | |
| **Summative and/or formative assessment** | |
| Summative only | **0 (0%)** |
| Formative only | **16 (25%)** |
| Summative and formative | **4 (7%)** |
| None | **43 (68%)** |
| **Types of outcomes** | |
| Health system and organization | **3 (5%)** |
| Collaboration and communication | **13 (21%)** |
| Patients' value and preferences | **8 (13%)** |
| Clinical practice and outcome | **9 (14%)** |
| Problem-solving skills | **2 (3%)** |
| Students' knowledge acquisition | **2 (3%)** |
| Satisfaction | **3 (5%)** |
| Students' professional development | **3 (5%)** |
| SDM behaviours | **1 (2%)** |
| Students' ethical reasoning decision | **2 (3%)** |
| Clinical decision-making skills | **7 (10%)** |
| End-of-life care | **5 (8%)** |
| Health professionals' attitude towards SDM | **1 (2%)** |
| Not reported | **4 (6%)** |

* Other countries: Brazil, Taiwan, Netherlands, Indonesia, Austria, Spain, Norway, Iran, France, Singapore, and Korea.

## 3. Results

Figure 1 summarizes the search results by using the PRISMA flow diagram template. We initially retrieved 3932 articles. Following removal of duplicates, we screened 516 articles for abstracts and removed 342. Of 174 articles, the full text was assessed for eligibility and 111 articles were excluded either because they failed to meet the population and intervention inclusion criteria, or the full article was unavailable (Table S1). We reviewed the

full text of the remaining 63 articles, and each of the included article scored seven or higher according to the quality assessment (Table S2).

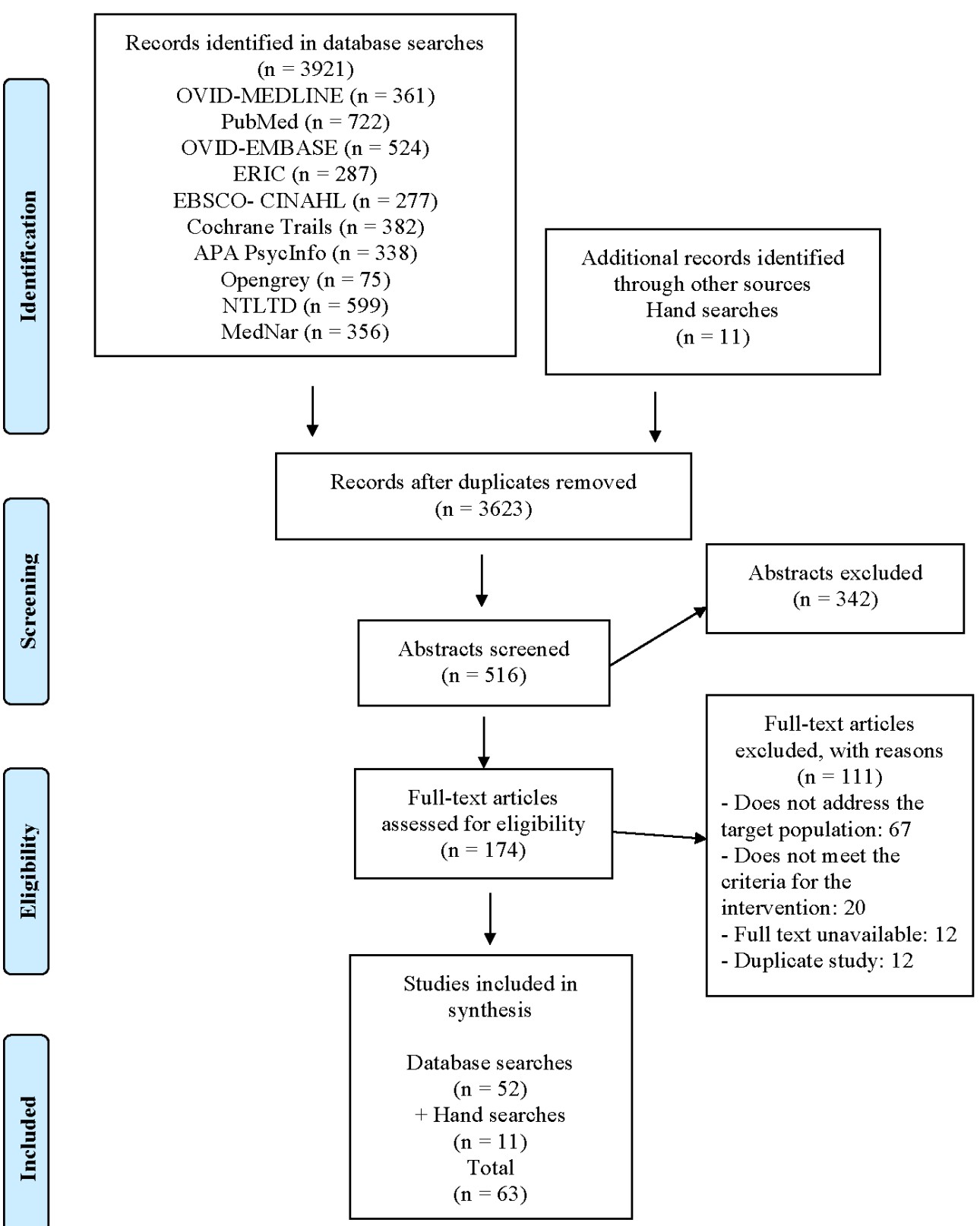

**Figure 1.** PRISMA flow diagram. Legend: The PRISMA diagram details our search and selection process applied during the scoping review.

### 3.1. Study Characteristics

Table 4 presents the general characteristics of studies. All articles were published between 2002 and 2020, with by far the majority (84%; 53/63) published after 2010 [13,18,21–71]. Most studies were carried out in Western countries, including 19 in Canada [12,21,22,29,31, 32,34,37,38,41,43,50,53,54,65,72,73], 16 in the USA [30,46–48,52,55–58,61–63,68,71,74,75], 9 in the UK [18,23,33,40,49,64,76–78], 4 in Australia [24,26,59,60], 4 in Germany [13,42,51,67], and 11 in other countries [25,27,28,35,36,39,44,45,66,69,70]. The mean length of the study period was approximately 8 months with a range of 7 days to 24 months, but 39 articles did not report the study period. Fifteen studies described instrument design, instrument validation, curriculum development, or curriculum design [18,28,29,33,36,40,57,58,62,68,71,73,75,78,79], and 14 studies were reviews [12,13,21,24–26,31,32,37,42,49,64,65,80]. Randomized controlled trials [41,50,52–54,63,66,69,78] and mixed-methods designs [30,35,38,39,44–46,67,72] were used in nine studies.

Table 1 also reports on the methodology of the studies. Review studies addressed the following topics: evaluating the effectiveness of SDM interventions [12,13,21,24,31,32,37], training on IP communication and SDM [26,64,65], and improving the quality of the healthcare system related to SDM [25,42,49,80]. Mixed-method designs were used in some studies to understand attitudes or intentions towards IP-SDM [35] and decision-making styles [30] and facilitate the development of an educational intervention [45]. Curriculum developments were addressed for primary healthcare [18,29], simulation settings [57,75], interprofessional teams supporting patients in healthcare decision making [58], and internal medicine for residents [71].

### 3.2. Theoretical Frameworks for IP-SDM Educational Interventions

#### 3.2.1. Educational Frameworks and Learning Theories

More than half of the studies (69%; 43/63) did not report using an educational framework or learning theory. Those that did (31%; 20/63) used adult learning theory [18,24,57,80], the Reflection in Clinical Decision-Making Revised Model [47,74], and experiential learning theory [27,73]. Each of the other examples are applied in one study: reflecting on learning [23], clinical decision-making model and Bloom's taxonomy [33], Kahneman model [36], constructivist learning theory [43], work-based experiential learning [77], Knowledge-to-Action Framework [53], model adapted from Braddock and colleagues [75], Kern's six-step approach to curriculum development [58], interprofessional healthcare team (IPHCT) meeting [59], social cognitive learning theory [64], the OncoTalk teaching model [68], and presage–process–product (3P) model [69].

#### 3.2.2. SDM Models and Their Components

More than half of the studies (58%; 37/63) did not report using SDM models. Examples of the studies that reported using SDM models (42%; 26/63) are categorized into communication and collaboration models: simple risk communication aids [78], three-talk model [61], Interprofessional Education for Collaborative Patient-Centred Practice (IECPCP) Synthesis Framework [77], and NephroTalk [68].

Models that help to make decisions include: IP-SDM model [29,45,53], Schon's model [47,74], transactional and descriptive model [12], SDM framework [21], Revised IP-SDM model [23], cardiopulmonary resuscitation (CPR) decision-making practices [24], TURF (Task, User, Representation, and Function) [33], Wilkinson's framework [39], DECISION+ [41], Ready for SDM [44], decision aids [42], computerized decision support systems (CDSS) [49], explanatory models of illness or decision-making [57], Outcome Present State model [64], Ottawa Decision Support Framework [65], DECISION+2 [66], model of integrated patient-centredness and expanded model of SDM [67], and "6 Steps to Shared Decision-Making" framework [71]. One model addresses conflict: O'Connor's Decisional Conflict Scale [52].

The shared decision-making models featured in the included studies have various components and may take different steps. CPR decision-making practices include: (i) knowing what to say; (ii) knowing how to say it; and (iii) wanting to say it [24]. The

interprofessional SDM (IP-SDM) model [29,45,53] has three levels: the individual (micro) level and two healthcare system (meso and macro) levels. DECISION+ has major and minor components related to participation in continuing professional development programmes in SDM [41]. Schon's model has both informative and directive factors that influence clinical decision making [47,74]. Interdisciplinary education processes and collaborative patient-centred practice are represented as separate components in the IECPCP Synthesis Framework [80].

### 3.3. IP-SDM Educational Applications and Delivery Methods

3.3.1. Population Characteristics

The studies included students (n = 1857), physicians (n = 901), allied healthcare professionals (n = 674), nurses (n = 126), and experts in SDM and IPE (n = 106). In total, 475 patients and caregivers were included [15,18,21–23,26–29,31–35,37–48,50,52–54,56–58,62–68,70–78,80]. The disease and medical specialties included internal medicine (34%; 22/63) [12,27,28,33,37,38, 41,43,47,48,50,53,54,61,62,66,68,71,74–76,78], end-of-life care and oncology (21%; 14/63) [21,24, 31,32,34,36,39,52,56,63,65,69,70,73], orthopaedic surgery (5%; 3/63) [22,23,77], traditional and complementary medicine (3 %; 2/63) [25,26], Down syndrome (2%; 1/63) [72], mental health (2%; 1/63) [42], and emergency medicine (2%; 1/63) [45]. Students involved in the studies were postgraduates (35%; 22/63) [28,34,37,38,41–43,45,50–54,56,61,66–68,70,71,77,78] or undergraduates (29%; 18/63) [23,24,27,30,33,35,36,39,40,55,59,62,63,69,73,75,76,80], and 11 studies included both (17%; 11/63) [22,26,29,31,44,47,48,58,64,65,74].

3.3.2. Intervention Characteristics

Interventions occurred in university and teaching clinics (26%; 17/63) [30,35,38–40, 44–46,55,58–60,62,63,66,70,76], hospital settings (15%; 10/63) [13,22,24,34,36,43,54,56,68,73], primary healthcare settings (14%; 9/63) [12,18,25,26,29,32,41,53,72], outpatient clinics (11%; 7/63) [23,28,42,51,67,71,74], and simulation settings (5%; 3/63) [27,69,75]. The mean duration of intervention was approximately 4 months with a range of <2 h to >12 months, but 43 articles did not report the intervention duration. Educational interventions focused on knowledge and skills (30%; 19/63) [18,24,27,28,30,37,40,41,43,44,49,53,59,63,68,73–75,79], knowledge, skills, and attitudes (27%; 17/63) [13,21,22,26,29,31,32,35,47,50,57,58,62,65,67,72, 80], skills only (16%; 10/63) [23,33,38,42,56,64,69,71,77,78], knowledge, skills, and attitudes (27%; 17/63) [13,21,22,26,29,31,32,35,47,50,57,58,62,65,67,72,80], skills only (16%; 10/63) [23, 33,38,42,56,64,69,71,77,78], knowledge and attitudes (8%; 5/63) [39,52,66,70,76], attitudes and skills (8%; 5/63) [12,35,45,46,61], and knowledge only (6%; 4/63) [36,48,51,55].

Teaching methods included workshops (17%; 11/63) [29,36,39,41,43,50,66,68,73,77,78], interactive learning sessions and discussions (15%; 9/63) [18,23,27,31,52,55,63,65,67], lectures (8%; 5/63) [35,44,58,59,63], case-based learning (8%; 5/63) [45,61,71,76,80], videos (6%; 4/63) [24, 52,53,72], role play (6%; 4/63) [13,28,57,62], observation (5%; 3/63) [47,48,74], simulation (5%; 3/63) [33,69,75], and online courses (5%; 3/63) [32,54,66]. Decision applications dealt with communication and collaboration (28%; 18/63) [22,24,27,31,35,41–43,45,49,54,59,62,64,67,69,73,78], patient care and satisfaction (16%; 10/63) [26,28,30,36,40,52,61,68,77,80], healthcare choice (13%; 8/63) [29,38,48,50,53,61,66,71], and decision quality (9%; 6/63) [12,23,52,56,65,75]. The data collection methods included questionnaires (47%; 29/63) [22,23,26–30,33,36,38,39,41,42,44,50–52,58,59,61–63,66,68–70,78,80], interviews (13%; 8/63) [12,40,45,47,48,72,74,77], focus groups (5%; 3/63) [24,43,67], and recorded discussions (5%; 3/63) [35,54,55].

Several studies described instrument design, instrument validation, and curriculum development and design. An example of a study that described instrument design is the Student's Inventory of Professionalism (SIP) including an SDM based on undergraduate education in palliative care [39]. Regarding instrument validation, two studies validated an IP-SDM model [23,28] by asking participants about proposed changes to the model, the potential barriers, and facilitators to the implementation of the model in clinical practice. The participants were also asked to assess the model using a theory appraisal questionnaire. Several studies addressed curriculum design and development, for example, a framework

utilized to develop a four-step intervention to improve advanced CPR decision making [27]. Other studies dealing with curriculum design are the TURF framework (Task, User, Representation, and Function) used to teach clinical reasoning and decision-making skills [33], and the modification of the six-step approach to curriculum by Kern et al. to improve continuing professional development for interprofessional teams supporting patients in a healthcare decision-making model [58]. A summer school programme for oncology comprised clinical and research parts to teach clinical decision making in a multidisciplinary environment [36]. Another is the Sim-IPE programme that conducts full-scale simulation and communication strategies adapted from Team STEPPS [69]. In addition, Fit for SDM is an example of a train-the-trainer programme conducted as a university project to teach staff about the healthcare team in terms of SDM [51]. NephroTalk is designed as a half-day workshop for dialysis decision making and end-of-life care in nephrology communication skills training for staff, patients, and family with chronic kidney diseases [68]. Another intervention is a workshop-based curriculum held for internal medicine residents to promote SDM education in treatment decisions [21].

### 3.4. Assessed Outcomes in IP-SDM Educational Interventions

#### 3.4.1. Evaluation Frameworks

Of the studies that reported using frameworks to evaluate IP-SDM outcomes (29%; 18/63), 6 studies applied Kirkpatrick's model [13,27,31,44,58,65] and 2 studies used Reflection-on-Action (ROA) [47,74]. Other assessment frameworks include the following: Evaluation by McDowell and Newell, and by Tremblay and collaborators [12], Flanagan's critical incident technique [34], integrated promoting action on research implementation in health services (iPARIHS) framework [43], signal detection theory [48], "PSA is a Decision" [52], CollaboRATE instrument [54], OPTION: observing patients, multilevel modelling involvement [78], Medical Research Council framework [79], SHARE (Seek participation, Help comparison, Assess values, Reach decision, Evaluate decision) [61], and Outcome Present State model [64].

#### 3.4.2. SDM Measures and Instruments

More than half of the studies (63%; 40/63) apply SDM measures and instruments. Examples are: theory appraisal questionnaire [72], 'Goals of Patient Care' (GOPC) form and Supportive and Palliative Care Indicators Tool (SPICT) tool [24], Assessment of Interprofessional Team Collaboration Scale (AITCS), Attitudes Toward Interprofessional Health Care Teams Scale (ATHCTS) [27], The Rational-Experiential Inventory (REI-40) [30], DECIDE quantitative [34], Readiness for Interprofessional Learning Scale (RIPLS) [35,59], compulsory pre-VSSO and post-VSSO single choice questionnaire [36], the OPTION 5 (observing patient involvement in decision-making) [38], validated OPTION-12 (O12) instrument [40,58,78] (Observing Patient Involvement) scale [40], DECISION+ and decision conflict scale [41], 9-item SDM Questionnaire (SDM-Q-9) [42], Cognitive Orientation to daily Occupational Performance (CO-OP) [43], TPB-based questionnaire [45], Interprofessional Collaborative Practice (IPCEP) Core Competency of Values/Ethics [46], Theory of Planned Behaviour based questionnaire [50], 3-item uncertainty subscale from O'Connor's Decisional Conflict Scale [52], patient questionnaires of validated scales—SPIRIT checklist [53], validated scoring tool for the degree of SDM [56], checklist on the elements of SDM [75], Team Dimensions Rating Form, Collaboration and Satisfaction About Care Decisions [58], Individual Teamwork Observation and Feedback Tool (iTOFT) [60], Team Oral Structured Clinical Examination or (TOSCE) [80], 12-item instrument that addressed students' communication skills [63], Decisional Conflict Scale [66], and end-of-life care performance scale [70] (Table 3).

#### 3.4.3. Type of Outcomes

Of all the studies, 94% mention types of outcome, most often collaboration and communication (21%; 13/63) [22,25,27,34,41,43–45,58–60,72,79], clinical practice and outcome (14%;

9/63) [18,23,53,54,64,65,68,70,71], patients' value and preferences (13%; 8/63) [21,24,28,29,38, 42,52,61], and clinical decision-making skills (10%; 7/63) [31,47,51,67,74–76]. Fewer studies assessed other outcomes, such as end-of-life care (8%; 5/63) [48,56,73,77,78], satisfaction (5%; 3/63) [32,63,80], students' professional development (5%; 3/63) [13,39,69], health system and organization (5%; 3/63) [12,37,49], problem-solving skills (3%; 2/63) [30,66], students' knowledge acquisition (3%; 2/63) [36,62], students' ethical reasoning decision (3%; 2/63) [46,55], SDM behaviours (2%; 1/63) [40], and health professionals' attitude towards SDM (2%; 1/63) [50].

### 3.4.4. Summative and Formative Assessments

Most of the articles did not have summative or formative assessments (68%; 43/63). Only some had a formative assessment (25%; 16/63) [23,27,28,36,38,40,50,60,62–64,69,71,73,74,78] or both summative and formative assessments (7%; 4/63) [13,44,58,75].

## 4. Discussion

This scoping review aimed to provide an extensive overview of the current knowledge regarding SDM interventions in health professions education. Our search was broad and targeted both published and unpublished articles. To reduce the risk of bias, we followed a strict methodology for screening articles and extracting data. We ultimately included 63 studies published mostly between 2002 and 2020 on theoretical frameworks used for IP-SDM educational interventions and their components (RQ1), current applications and delivery methods of IP-SDM educational interventions (RQ2), and outcomes assessed in IP-SDM educational interventions (RQ3). This review reveals the diversity of approaches to IP-SDM in health professions education in interventions occurring in North America, Australia, and Europe. Very few reported interventions took place in other countries, which could be due to the inclusion criteria of articles in the English language. The interventions varied in duration, clinical setting, health professionals' involvement, patient and family members' involvement, as well as in the use of educational frameworks, SDM models, and evaluation frameworks. This heterogeneity makes it difficult to compare the results of the studies included in the review.

Regarding RQ1 (theoretical frameworks for IP-SDM educational interventions and their components), only one-third (31%) of the included studies reported on educational frameworks and learning theories, while not even half of them (42%) reported on SDM models. As SDM is a broad area, little information was addressed about how to implement SDM interventions [14]. Yet, the focus on interprofessional collaboration is increasing in healthcare research, since SDM is applied in many settings, including university and teaching clinics, hospital settings, primary healthcare settings, outpatient clinics, and simulation settings. Neither the theoretical framework nor SDM models were frequently reported, and if they were, the diversity was huge. There was no leading theoretical framework, and the IP component was seldom mentioned in SDM models. This shows how broad the field of IP-SDM is but makes it difficult to compare studies. Furthermore, most of the SDM models, tools, and designs were developed for a particular study and lacked evidence of validity and reliability. Thus, there is a need to address frameworks and outcomes to assess the effectiveness of IP-SDM interventions for health professions education.

Studies relevant to RQ2 (applications and delivery methods of IP-SDM) reported using multiple active teaching methods to engage students in the process of gaining knowledge, skills, and attitude, such as videos, role play, interactive lectures, case-based learning, online courses, blended learning, simulation sessions, and workshops. Students' active engagement positively affects their learning outcomes in clinical practice [81]. SDM interventions were mainly targeted to medical students and fewer other health professions students such as nurses, pharmacists, and allied healthcare professionals. This could be due to the great interaction between patients and physicians in clinical practice and the power of physicians in decision making [11]. Medical students involved in interventions included almost 35% on the postgraduate level. Few programmes targeted the undergraduate level because of the complex communication and clinical skills needed in SDM [82]. Healthcare

receivers were primarily patients under internal medicine, orthopaedic, and end-of-life care, which requires interprofessional collaboration among HCPs and decision making in these specialties. Engaging patients and their family members in the SDM process in clinical practice is crucial [83]. This review identified several types of decision and applications that concern quality of patient decision, patient care, satisfaction, communication, and collaboration. This underlines the need to include patients and their family members in SDM in health professional teaching activities [84].

The studies relevant to RQ3 (outcomes assessed in IP-SDM educational interventions) involved 18 evaluation frameworks, of which 6 applied Kirkpatrick's model. Very few interventions were based on a summative and formative assessment of the learning, although we identified a variety of evaluation frameworks. As IP-SDM involves teamwork, which is difficult to assess for specific student performances [85], SDM interventions should be based on learning theories and educational frameworks and should be evaluated with reliable and valid measurement tools to enhance teaching effectiveness [86]. Longitudinal study application should be considered in such interventions. IP-SDM education should be encouraged for all HCPs to ensure a better impact on SDM in clinical practice.

## 5. Limitations of This Scoping Review

This review is limited to the years 2000–2020. Articles published before 2000 that might have retained relevance were excluded. Non-English articles were also excluded and so we might have missed relevant articles published in other languages.

Our review identifies heterogeneity among studies in terms of the study population, educational interventions, and measured outcomes. As SDM varies across countries and implicitly implies the involvement of multiple people and professionals who make the decisions, there is an inevitable lack of explicit IP components. This means that the results of this review cannot be generalized.

## 6. Conclusions

The objective of this review was to provide an overview of current IP-SDM educational interventions with respect to their theoretical frameworks, delivery, and outcomes in healthcare settings. The articles included in the review demonstrate interest in teaching IP-SDM knowledge, skills, and attitudes in health professions education. This overview of current trends highlights the use of active educational methods and the need to involve patients and their family members in the educational activity. The identified educational interventions varied in terms of health professionals' involvement, intervention duration, educational frameworks, SDM models, and evaluation frameworks. Using theoretical frameworks for learning, assessment, and evaluation of the IP-SDM intervention is recommended for developing a curriculum to teach IP-SDM to healthcare professions students. In the review, we suggested the need for more homogeneity in theoretical frameworks and validated measures to assess IP-SDM.

## 7. Practice Implications

Our scoping review revealed considerable interest in IP-SDM in health professions education. We found several educational interventions targeting HCPs in undergraduate and postgraduate studies, but these were heterogeneous in terms of health professionals' involvement, intervention duration, educational frameworks, SDM models, and evaluation of frameworks and outcomes. It is therefore difficult to compare the design and delivery of IP-SDM in health professions education. As many health professionals are expected to have the necessary knowledge, attitudes, and skills related IP-SDM in healthcare, we think there is a need for a framework for the development, teaching, and assessment of IP-SDM based on evidence and theory. It could start in undergraduate education not too early and not too late, and to be continued on the postgraduate level so that future HCPs become better equipped to deal with the care needs of patients and their family members.

HCP educators should prepare educational activities that contribute to improving patients' outcomes for a better healthcare delivery.

## 8. Lessons for Practice

- More than half of the studies did not report using an educational framework or learning theory or SDM models. The one who did had various components and different steps. The studies that reported using SDM models are focused on communication and collaboration or decision aids.
- The current delivery methods of IP-SDM educational intervention included workshops, interactive learning sessions, case-based learning, videos, role play, observation, simulation, and online courses.
- The outcomes of IP-SDM educational interventions included collaboration and communication, clinical practice and outcome, patients' value and preferences, and clinical decision-making skills.

**Supplementary Materials:** The following supporting information can be downloaded at: https://www.mdpi.com/article/10.3390/su142013157/s1, Table S1: Excluded articles; Table S2: Quality assessment of included articles.

**Author Contributions:** Study conception and design: L.S., B.A., N.D.J. and J.D.N.; analysis and interpretation of results: L.S. and B.A.; draft manuscript preparation: L.S. All authors have read and agreed to the published version of the manuscript.

**Funding:** This research received no external funding.

**Institutional Review Board Statement:** Not applicable.

**Informed Consent Statement:** Not applicable.

**Data Availability Statement:** Not applicable.

**Conflicts of Interest:** The authors declare no conflict of interest.

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
