# Peer review of "Current Trends in Interprofessional Shared Decision-Making Programmes in Health Professions Education: A Scoping Review"

_sustainability, doi:10.3390/su142013157_

Round 1
Reviewer 1 Report
The text presented complies with what is required for a review of the literature and following the PRISMA method, the authors have identified relevant documents for analysis.
Despite this, I consider that some changes are necessary, which I detail below:
the articles that are part of the analysis should not appear cited in the theoretical framework.
The number of references used in the framework is scarce.
Review articles should not be part of the analysis, to avoid duplication.
The Results table should be presented following a chronological order, to perceive more clearly the advances.
in the References, it is necessary to review in detail, for example, the page numbers.
there is additional information in the form of annexes that I consider not necessary to publish. the extension far exceeds what is expected in a scientific article.
Author Response
|
Reviewer comment |
Author response/modification/change |
|
Reviewer 1 |
|
|
The text presented complies with what is required for a review of the literature and following the PRISMA method, the authors have identified relevant documents for analysis. Despite this, I consider that some changes are necessary, which I detail below: |
Thank you for your comment. We will take your points into consideration. |
|
the articles that are part of the analysis should not appear cited in the theoretical framework. |
Could you please clarify this comment? The articles need to be cited in the theoretical framework part for the readers references and answering the RQ1: What are the components of IP-SDM educational interventions and which theories are they based on? |
|
The number of references used in the framework is scarce. |
The reason behind this is that more than half of the studies (69%; 43/63) did not report using an educational framework or learning theory. Moreover, more than half of the studies (58%; 37/63) did not report using SDM models. |
|
Review articles should not be part of the analysis, to avoid duplication. |
Review articles were included in the eligibility criteria in line 159 . To avoid duplication tables’ contents of 3,4 & 5 were minimized. |
|
The Results table should be presented following a chronological order, to perceive more clearly the advances. |
Totally agree and modification was done accordingly for tables 3,4 & 5. |
|
in the References, it is necessary to review in detail, for example, the page numbers. |
Thank you for the suggestions. In the tables 3,4 & 5 you can find the important points for each reference. |
|
there is additional information in the form of annexes that I consider not necessary to publish. the extension far exceeds what is expected in a scientific article. |
Totally agree and modification was done accordingly. Table contents were minimized. Is that enough or would you suggest another thing?
|
Reviewer 2 Report
These comments are mostly minor for the authors reflection and consideration.
After reading the abstract I was interested to know if any papers measured students’ feelings of confidence or competence – were there any objective measures? Would be useful to also point to ideas for future research in the abstract. It would also be useful to mention which discipline most of the articles came from – medicine? Nursing? Allied health? How many were undergraduate and how many were postgraduate studies?
The results of each of the 3 research questions should be reported in the abstract. For example, the finding that more than half of the studies (69%) did not report a theoretical framework is very interesting and important yet is not mentioned at all in the abstract.
Well written introduction, concise, identifying the gaps in previous literature plus highlighting the rationale and importance of the study
Clear expression of RQs
Methods
Was any particular software used to assist with the title and abstract screening process? E.g Covidence?
Quality assessment – line 172 – should say ‘developed by Buckley’ et al bit ‘conform’
How was consensus reached if there was disagreement by the raters about inclusion or exclusion criteria in selecting abstracts and articles for the full review?
Results
In line 220 it says the mean length of the study time/intervention duration was 8 months but in the abstract the authors report 4 months. Which one is it?
Overall the results and analysis were interesting to read and revealing about the state of research in this space which needs a higher level organisation framework to assist comparison. Great points were made in the discussion which need to also be articulated in the abstract too.
Table 3
A lot of work went into collating this information. One column – the definition of SDM – could have been summarized in the reporting of results. A qualitative or narrative synthesis of these types of definitions alone would be interesting to read in the results section.
As the research you reviewed spans 20 years it would have been interesting to see what trends were observed over time – what were the most common research methods/evaluations/focus points for research post 2010 compared to pre 2010 for example?
Author Response
|
Reviewer comment |
Author response/modification/change |
|
Reviewer 2
|
|
|
These comments are mostly minor for the authors reflection and consideration. |
Thank you for your comment. |
|
After reading the abstract I was interested to know if any papers measured students’ feelings of confidence or competence – were there any objective measures? |
We did not find that in the papers. |
|
Would be useful to also point to ideas for future research in the abstract. |
Practice implications: We need more homogeneity in both theoretical frameworks and validated measures to assess IP-SDM. |
|
It would also be useful to mention which discipline most of the articles came from – medicine? Nursing? Allied health? |
We included the numbers of participants from each discipline in 3.3.1 population characteristics paragraph in details. |
|
How many were undergraduate and how many were postgraduate studies? |
Totally agree and modification was done accordingly. This is added in the abstract in line 25 & 26. |
|
The results of each of the 3 research questions should be reported in the abstract. For example, the finding that more than half of the studies (69%) did not report a theoretical framework is very interesting and important yet is not mentioned at all in the abstract. |
Totally agree and modification was done accordingly. The abstract now includes the following sentence. |
|
Well written introduction, concise, identifying the gaps in previous literature plus highlighting the rationale and importance of the study |
Thank you for your comment. |
|
Clear expression of RQs |
Thank you for your comment. |
|
Methods Was any particular software used to assist with the title and abstract screening process? E.g Covidence? |
No, we used Microsoft Excel. |
|
Quality assessment – line 172 – should say ‘developed by Buckley’ et al bit ‘conform’ |
Totally agree and modification was done accordingly in line 173. |
|
How was consensus reached if there was disagreement by the raters about inclusion or exclusion criteria in selecting abstracts and articles for the full review? |
We haven’t faced disagreement. |
|
Results In line 220 it says the mean length of the study time/intervention duration was 8 months but in the abstract the authors report 4 months. Which one is it?
|
There is a difference between study period and intervention duration of the educational activity. In the 3.1. study characteristics paragraph, it was mentioned that: “The mean length of the study period was approximately 8 months.” In the 3.3.2. Intervention characteristics paragraph, it was mentioned that: “The mean duration of intervention was approximately 4 months”. |
|
Overall the results and analysis were interesting to read and revealing about the state of research in this space which needs a higher level organization framework to assist comparison. |
Thank you for your comment. In the conclusion, it was mentioned that “However, the identified educational interventions were heterogeneous in health professionals' involvement, intervention duration, educational frameworks, SDM models, and evaluation frameworks”. Which makes it difficult to compare. |
|
Great points were made in the discussion which need to also be articulated in the abstract too |
Thank you for your comment. As per Instructions for Authors in the abstract section should follow the style of structured abstracts including: Background, Methods, Results & Conclusion. |
|
Table 3
A lot of work went into collating this information. One column – the definition of SDM – could have been summarized in the reporting of results. A qualitative or narrative synthesis of these types of definitions alone would be interesting to read in the results section. |
Totally agree with the reviewer and thank you for this point, but that is beyond the scope of this review.
|
|
As the research you reviewed spans 20 years it would have been interesting to see what trends were observed over time – what were the most common research methods/evaluations/focus points for research post 2010 compared to pre 2010 for example? |
Thank you for your suggestion. It will be considered in further reviews. For the meantime, we were focusing on the results part to answer the 3 RQs: Research Question 1 (RQ1): What are the components of IP-SDM educational interventions and which theories are they based on?
Research Question 2 (RQ2): What are the current delivery methods of IP-SDM educational interventions?
Research Question 3 (RQ3): What are the outcomes of IP-SDM educational interventions and how are these assessed? |
Reviewer 3 Report
Dear authors,
The study presented provides an interesting approach to a subject that has not been studied, therefore congratulations on the subject decision.
When reading the manuscript, concepts are being presented and too extensively referenced. The extended referencing with over 8 references makes it a challenge to read and follow the lead. That could be revised and improved.
Best regards
Author Response
|
Reviewer comment |
Author response/modification/change |
|
Reviewer 3
|
|
|
The study presented provides an interesting approach to a subject that has not been studied, therefore congratulations on the subject decision. |
Thank you for your comment. |
|
When reading the manuscript, concepts are being presented and too extensively referenced. The extended referencing with over 8 references makes it a challenge to read and follow the lead. That could be revised and improved. |
Thank you for your point. However, we referred to the references for readers to identify and locate the work cited.
|
Reviewer 4 Report
This is a comprehensive report on IP-SDM in the literature. The scope is extensive and the report detailed.
The text has a number of grammatical errors some which are highlighted in red.
The review is rather descriptive and lacks the depth of synthesis it deserved. It would benefit from summarising authors' take on the 3 questions of the report by putting together the results of the literature reviewed and come up with a set of recommendations for methodologies and interventions used based on theoretical models. The outcome of validated interventions could have been more closely looked at conclusions drawn.
Given the importance of this study, it is uncertain why the grey literature was included.
Lastly, the review includes a number of systematic reviews. It is not clear how these systematic reviews were evaluated, as each is a review of a number of included literature and interventions evidenced.
See highlighted areas in the attached file.

Author Response
|
Reviewer comment |
Author response/modification/change |
|
Reviewer 4
|
|
|
This is a comprehensive report on IP-SDM in the literature. The scope is extensive and the report detailed. |
Thank you for your comment. |
|
The text has a number of grammatical errors some which are highlighted in red. |
Modification was done accordingly. |
|
The review is rather descriptive and lacks the depth of synthesis it deserved. It would benefit from summarising authors' take on the 3 questions of the report by putting together the results of the literature reviewed and come up with a set of recommendations for methodologies and interventions used based on theoretical models. The outcome of validated interventions could have been more closely looked at conclusions drawn. |
Thank you for your comment. We cannot give that at the moment, because due to the heterogeneity of the results, we are not able to draw firm conclusions on what works in what context or how that should be measured. Therefore, we need first more homogeneity in and consensus about both theoretical frameworks and validated measures. |
|
Given the importance of this study, it is uncertain why the grey literature was included. |
Since the article was a scoping review, grey literature was included. |
|
Lastly, the review includes a number of systematic reviews. It is not clear how these systematic reviews were evaluated, as each is a review of a number of included literature and interventions evidenced. See highlighted areas in the attached file. |
The review was evaluated as a whole. The quality of each article that met the study inclusion criteria was assessed with 11 quality indicators for selection developed by Buckley et al. |